# QHyer: Q-conditioned Hybrid Attention-Mamba Transformer for Offline Goal-conditioned RL

Xing Lei [1]  Jincheng Wang [2]  Xuetao Zhang [1]  Donglin Wang [3]

## Abstract

Offline goal-conditioned RL (GCRL) learns goal-reaching policies from static datasets, but real-world environments are often partially observable, so the collected trajectories are only partly consistent with the Markov assumption while other segments remain history-dependent. History-aware sequence models such as Decision Transformer (DT) are a natural fit for long-term dependency modeling, yet pure attention is inefficient and brittle when handling local Markovian structure and long-range context simultaneously. Although recent hybrid architectures (e.g., LSDT) introduce local extractors, their fixed-window extraction cannot adapt the effective memory to varying dependency lengths, often truncating long-range context instead of compressing it. Moreover, under sparse rewards, return-to-go (RTG) becomes non-discriminative across sub-trajectories, offering little guidance for stitching goal-reaching behaviors from diverse demonstrations. To address these limitations, we propose **QHyer** (**Q**-conditioned **Hy**brid Attention-Mamba Transform**er**), which replaces RTG with a Normalizing Flows (NFs) parameterized goal-reaching Q-estimator used directly as conditioning tokens, and a gated Hybrid Attention-Mamba backbone whose selective state-space dynamics enable content-adaptive history compression while attention captures global goal-directed dependencies. Extensive experiments on OGBench and D4RL demonstrate that **QHyer** achieves state-of-the-art performance on both non-Markovian and Markovian datasets, validating its effectiveness for diverse scenarios.

[1]State Key Laboratory of Human-Machine Hybrid Augmented Intelligence, Institute of Artificial Intelligence and Robotics, Xi'an Jiaotong University [2]Computer Science, University College London [3]School of Engineering, Westlake University. Correspondence to: Xuetao Zhang <xuetaozhang@xjtu.edu.cn>.

*Proceedings of the $43^{rd}$ International Conference on Machine Learning*, Seoul, South Korea. PMLR 306, 2026. Copyright 2026 by the author(s).

## 1. Introduction

Offline Goal-Conditioned Reinforcement Learning (Offline GCRL) aims to learn goal-reaching policies from static datasets, offering a promising paradigm for real-world applications where online interaction is costly or infeasible (Levine et al., 2020; Liu et al., 2022). While most existing offline GCRL datasets are collected by Markovian behavior policies, an increasing number of practical datasets exhibit non-Markovian properties where actions depend on historical context rather than current observations alone (Park et al., 2025). These properties pose fundamental challenges for existing value-based methods (Kostrikov et al., 2022; Park et al., 2023; Zhou & Kao, 2026; Ahn et al., 2026; Giammarino et al., 2026; Giammarino & Qureshi, 2026) that rely on Bellman backup. In contrast, sequence modeling approaches such as Decision Transformer (DT) (Chen et al., 2021) are naturally suited to non-Markovian settings by conditioning on return-to-go (RTG), states, and actions, leveraging self-attention to capture long-range dependencies from extended historical sequences.

Although DT naturally handles non-Markovian patterns through history conditioning, it exhibits two fundamental limitations when applied to offline GCRL. First, RTG is *trajectory-dependent* rather than *state-action-dependent*: it assigns values based on trajectory success rather than the quality of individual state-action pairs, providing *no discriminative signal* for distinguishing promising state-action pairs within failed trajectories. This is a critical requirement for trajectory stitching under goal-conditioned sparse rewards. Second, pure attention struggles to efficiently balance global goal-directed reasoning with fine-grained local dynamics modeling. Recent hybrid architectures such as LSDT (Wang et al., 2025) and DMixer (Zheng et al., 2025) incorporate convolution alongside attention to capture local patterns. However, non-Markovian offline GCRL data exhibits variable-length temporal dependencies that change dynamically across states and trajectory segments. Convolution with fixed receptive fields either wastes model capacity on irrelevant context when dependencies are short, or truncates critical information when dependencies are long, unable to adapt to this inherent variability.

We propose **QHyer** (**Q**-conditioned **Hy**brid Attention-

Mamba Transform**er**), the first sequence modeling framework to *jointly* resolve both limitations for offline GCRL. Our key observation is that these two limitations are *coupled*: effective trajectory stitching under sparse rewards requires both a state-action-dependent value signal *and* an architecture whose effective memory matches the temporal structure of the underlying behavior policy. Addressing either limitation in isolation is insufficient, since Q-conditioning layered on a fixed-window hybrid backbone inherits the convolutional pathology on non-Markovian `play` data, while improving the temporal architecture while retaining RTG still suffers from the trajectory-dependence bottleneck. Concretely, we (i) replace trajectory-dependent RTG with state-action-dependent Q-values estimated via Normalizing Flows (NFs) (Ghugare & Eysenbach, 2026), chosen specifically for their exact, properly normalized log-density, a property CVAEs, contrastive critics, and diffusion likelihoods cannot provide (Section 3.1), and (ii) design a gated Hybrid Attention-Mamba (Gu & Dao, 2024) backbone where Mamba's input-dependent selective state-space dynamics provide content-adaptive history compression, adjusting effective memory per-token rather than through a hand-tuned receptive field. Unlike prior value-guided Decision Transformer variants (Yamagata et al., 2023; Wang et al., 2024; Hu et al., 2024; Zhuang et al., 2024; Zheng et al., 2026), which *retain* RTG and attach Q-values as auxiliary losses or regularizers, **QHyer** *eliminates* RTG and uses Q-values directly as conditioning tokens. Under goal-conditioned sparse rewards, where RTG collapses to a near-binary signal, this distinction is decisive (Figures 2 and 5).

Our evaluation on OGBench (Park et al., 2025) and D4RL (Fu et al., 2020) demonstrates that **QHyer** achieves state-of-the-art performance across both non-Markovian datasets (OGBench `play` and D4RL `Maze`) and Markovian datasets (OGBench `noisy`), validating the effectiveness of NFs-based Q-value conditioning and the Hybrid Attention-Mamba architecture for offline GCRL.

## 2. Background

### 2.1. Offline GCRL

Offline GCRL is defined over a Markov Decision Process (MDP) $(\mathcal{S}, \mathcal{A}, \mathcal{G}, p, \gamma)$, where $\mathcal{S}$ denotes the state space, $\mathcal{A}$ the action space, $\mathcal{G}$ the goal space, $p(s'|s,a)$ the transition dynamics, and $\gamma \in [0,1)$ the discount factor. Following prior work (Park et al., 2023; 2025), we assume $\mathcal{G} \equiv \mathcal{S}$. The agent has access only to a static dataset $\mathcal{D} = \{\tau_i\}_{i=1}^N$ collected by behavioral policies $\beta$, where each trajectory takes the form $\tau^{(i)} = \{(s_t, a_t, s_{t+1})\}_{t=0}^{T^{(i)}-1}$. The objective is to learn a goal-conditioned policy $\pi(a|s,g)$ that maximizes the expected cumulative return $J(\pi) =$ $\mathbb{E}_{\tau \sim p^\pi, g \sim p(g)}[\sum_{t=0}^T \gamma^t r(s_t, g)]$ without interaction in the environment. To obtain goal-conditioned supervision, we employ hindsight experience replay (HER) (Andrychowicz et al., 2017), which samples goals from future achieved states along the same trajectory.

In standard GCRL with sparse rewards, the reward function is defined as $r(s, g) = \mathbb{1}[s = g]$, where the agent receives $1$ only upon reaching the goal and $0$ otherwise. Consequently, most state-action pairs yield no learning signal for states far from the goal. To address this, following prior work (Eysenbach et al., 2021; 2022), a probabilistic reward can be defined as $r(s, a, g) \triangleq (1 - \gamma) \cdot p(g|s')$, where $s'$ is the next state. Under this formulation, the goal-conditioned Q-function corresponds to the discounted state occupancy measure (Eysenbach et al., 2022; Bortkiewicz et al., 2025):

$$Q^\pi(s, a, g) = p_+^\pi(s^+ = g|s_0 = s, a_0 = a)$$
$$\triangleq (1 - \gamma) \sum_{k=0}^\infty \gamma^k p^\pi(s_{k+1} = g|s_0 = s, a_0 = a), \tag{1}$$

where $s^+ \triangleq s_K$ for $K \sim \text{Geom}(1 - \gamma)$ denotes a future state sampled at a geometrically distributed time step. Unlike sparse rewards that are directly observed, this formulation requires learning a density model to estimate Q-values.

### 2.2. NFs for Q-Value Estimation

NFs (Zhai et al., 2025) are invertible generative models that learn a bijective mapping $f_\theta : \mathbb{R}^d \to \mathbb{R}^d$ from a complex data distribution to a simple prior $p_0$ (typically standard Gaussian), with density computed exactly via the change of variables formula:

$$p_\theta(x) = p_0(f_\theta(x)) \left| \det \frac{\partial f_\theta(x)}{\partial x} \right|. \tag{2}$$

Following Ghugare & Eysenbach (2026), NFs can be constructed using coupling layers (Dinh et al., 2017). For the $t$-th block with input $x^t$ and condition $y$:

$$x_1^t, x_2^t = \text{split}(x^t),$$
$$\tilde{x}_2^t = (x_2^t + a_\theta^t(x_1^t, y)) \times \exp(-s_\theta^t(x_1^t, y)),$$
$$\tilde{x}^t = \text{concat}(x_1^t, \tilde{x}_2^t), \tag{3}$$

where $\text{split}(\cdot)$ partitions the input into two halves along the feature dimension, $a_\theta^t$ and $s_\theta^t$ are neural networks that output translation and scale parameters respectively.

In Offline GCRL, NFs can directly estimate it by modeling $p_\theta(g|s, a)$ (Ghugare & Eysenbach, 2026). The conditioning information $(s, a)$ is encoded by a state-action encoder, and the behavior Monte Carlo (MC) Q-value is obtained as:

$$Q_\theta^\beta(s, a, g) = \log p_0(f_\theta(g; z)) + \log \left| \det \frac{\partial f_\theta(g; z)}{\partial g} \right|, \tag{4}$$

where $f_\theta(\cdot; z)$ is the conditional NFs mapping goals to the latent space with $z$ being the encoded state-action representation. Note that $Q_\theta^\beta(s, a, g) = \log p_\theta(g \mid s, a)$ is an *exact, properly normalized* conditional log-density, which we use directly as the conditioning signal. During inference, we use $\exp(Q_\theta^\beta)$ when probability interpretation is needed.

In practice, the NFs model is trained via maximum likelihood on hindsight-relabeled transitions:

$$\mathcal{L}_{\text{NFs}} = -\mathbb{E}_{(s_t, a_t, g) \sim \mathcal{D}} \left[ \log p_\theta(g | s_t, a_t) \right]. \quad (5)$$

### 2.3. Sequence Modeling for Decision Making

Decision Transformer (DT) (Chen et al., 2021) models decision-making from offline datasets as a sequence modeling problem. Unlike traditional RL methods that estimate Q-functions or compute policy gradients, DT generates an action $a_t$ at timestep $t$ conditioned on the context of the previous $K$ timesteps along with the current state and return-to-go (RTG). The input sequence is formulated as $\tau = (\hat{R}_{t-K+1}, s_{t-K+1}, a_{t-K+1}, \ldots, \hat{R}_{t-1}, s_{t-1}, a_{t-1}, \hat{R}_t, s_t)$, where RTG $\hat{R}_t = \sum_{t'=t}^{T} r_{t'}$ is defined as the sum of rewards from the current step to the end of the trajectory and $K$ is the context length. For each timestep, three tokens (RTG, state, and action) are embedded and fed into the model. DT employs a causal Transformer that leverages self-attention layers to capture long-range dependencies.

Decision Mamba (DMamba) (Ota, 2024) integrates the Mamba (Gu & Dao, 2024) architecture into the DT framework by replacing self-attention with the Mamba block. The DMamba block first applies a one-dimensional causal convolution to extract local features:

$$x' = \text{SiLU}(\text{Conv1d}(x)), \quad (6)$$

where Conv1d operates with a local kernel over adjacent positions. The transformed sequence is then processed by the discrete-time selective state space model (SSM):

$$h_t = \bar{A} h_{t-1} + \bar{B} x'_t, \quad y_t = C h_t, \quad (7)$$

where $h_t$ is the hidden state and $y_t$ is the output. The key innovation of Mamba is the input-dependent selective mechanism:

$$B = \text{Linear}_B(x'), \quad C = \text{Linear}_C(x'),$$
$$\Delta = \text{softplus}(\text{Linear}_\Delta(x')), \quad (8)$$

where $\Delta$ controls the discretization step size.

## 3. QHyer: Unlocking Sequence Modeling for Offline GCRL

While sequence modeling naturally addresses the non-Markovian challenge, DT-based methods exhibit critical limitations when applied to GCRL. We propose **QHyer**, which

introduces NFs-based Q-value conditioning (Section 3.1) and a Hybrid Attention-Mamba architecture (Section 3.2) to overcome these limitations. The overall architecture is illustrated in Figure 1.

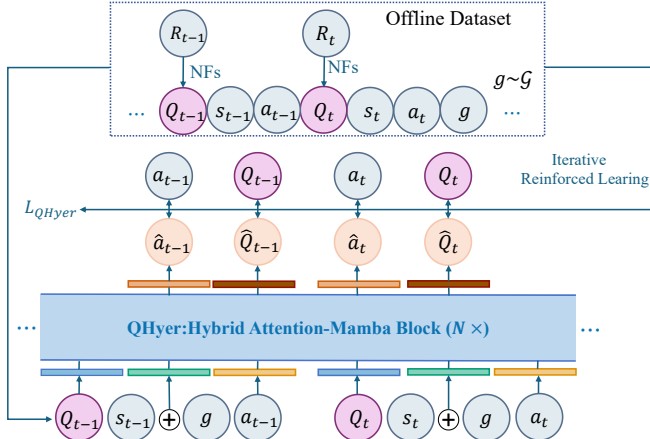

*Figure 1.* **Overview of QHyer architecture.** The framework consists of three main components: (1) a NFs Q-value estimator that replaces RTG conditioning, (2) a Hybrid Attention-Mamba Block, (3) concatenated state-goal tokenization for effective goal information propagation and (4) reinforced learning with expectile regression.

### 3.1. Limitation 1: RTG Fails Under Sparse Rewards

In standard DT-based methods, the return-to-go (RTG) serves as the conditioning signal that guides action generation. However, RTG is fundamentally inadequate for Offline GCRL with sparse binary rewards.

**The Root Cause: Trajectory-Dependence Prevents Stitching.** The fundamental limitation of RTG lies in its *trajectory-dependence*: RTG answers "did this trajectory succeed?" rather than "how valuable is this state for reaching the goal?" Consider a state $s$ that appears on both a successful trajectory (RTG=1) and a failed one (RTG=0). RTG assigns contradictory values to the *same state* based solely on trajectory outcome, making cross-trajectory comparison impossible. This directly prevents trajectory stitching because composing segments from different trajectories requires a *trajectory-agnostic* value metric that RTG fundamentally cannot provide. As shown in Figure 2 (a) (b), successful and failed trajectories receive uniformly different RTG values regardless of state quality, with only 25% of state-action pairs receiving discriminative signals.

**Our Key Insight: From Trajectory-Dependence to State-Dependence.** The Q-function $Q^\beta(s, a, g) = p_+^\beta(g|s, a)$ represents the probability of reaching goal $g$ from state-action pair $(s, a)$, measured *independently* of which trajectory that pair came from. This *state-dependence* enables a fundamentally new capability: identifying high-value segments from failed trajectories (they have high $Q^\beta$ despite low RTG) and composing them toward goals. Figure 2 (c) confirms this

| (a) Trajectories | (b) RTG | (c) $Q^\beta$ | (d) High-Q Segments |
|---|---|---|---|
| 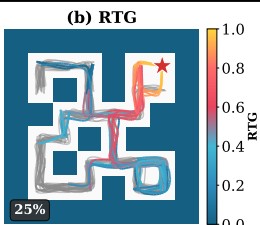 | 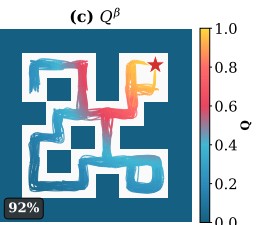 | 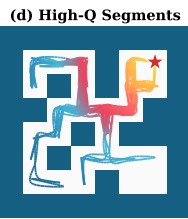 | |

*Figure 2.* **RTG vs. NFs-based Q-value conditioning in D4RL non-Markovian AntMaze-medium dataset.** (a) Trajectories: successful (blue) and failed (purple). (b) RTG conditioning: successful trajectories show color gradient while failed trajectories are uniformly gray (no signal), yielding poor coverage. (c) NFs-based $Q^\beta$ conditioning: all trajectories colored by Q-value, achieving significantly better coverage. (d) High-Q segments: trajectory portions where $Q^\beta$ exceeds a threshold naturally form paths toward the goal, enabling trajectory stitching even from failed demonstrations. Coverage percentages indicate the fraction of state-action pairs whose conditioning signal varies non-trivially (variance $> 0.01$) across trajectory segments.

prediction, showing that Q-value conditioning achieves 92% coverage compared to RTG's 25%. Figure 2 (d) further illustrates that high-Q segments naturally form paths toward goals even when extracted from failed demonstrations.

**Why MC Estimation Instead of TD Learning.** Having established the need for Q-value conditioning, we must choose how to estimate Q-values. Many standard offline RL methods (Fujimoto & Gu, 2021; Kostrikov et al., 2022) are built upon temporal difference (TD) learning. While TD learning can learn optimal value functions and possesses stitching capability, its reliance on bootstrapping leads to compounding errors that hinder the acquisition of optimal policies, particularly in long-horizon tasks (Myers et al., 2026; Park et al., 2026b). In contrast, MC learning directly estimates the cumulative reward for reaching a goal. By integrating it with a maximum Q-expectile regression loss proposed in our later analysis, we theoretically demonstrate that our method can also converge to an optimal stitched policy. Empirically, recent MC-based contrastive RL approaches (Eysenbach et al., 2022; Myers et al., 2026) have been shown to consistently outperform TD-based methods on long-horizon GCRL tasks.

**Why NFs for MC Q-Estimation.** Given that MC estimation is preferable, we must choose how to model the Q-value density $p_+^\beta(g \mid s, a)$. Our framework places one structural requirement on this density model. It must produce an *exact, properly normalized* log-density. The expectile target in Equation (10) is defined on $\log p_\theta(g \mid s, a)$ directly, and the transformer consumes Q-tokens that span *multiple goals* within one context window (Section 3.3), so goal-independent normalization is necessary for the learned Q-to-action pattern to transfer across goals. This requirement rules out the otherwise reasonable alternatives.

Conditional VAEs (Sohn et al., 2015) produce only the ELBO, a structural lower bound that cannot be closed by increasing capacity, and which distorts the Q-landscape in a goal-dependent way. Contrastive RL (Eysenbach et al., 2022) trains a binary cross-entropy classifier whose Bayes-optimal output is the log density ratio $\log \frac{p(g|s,a)}{p(g)}$. While the goal-dependent partition $p(g)$ cancels when selecting ac-

tions at a *fixed* goal, it introduces goal-dependent offsets in the Q-token sequence our transformer reads across *multiple* goals, which degrades cross-goal conditioning. Diffusion models (Ho et al., 2020) and continuous flow-matching objectives (Lipman et al., 2023) can reach high sample quality, but their per-sample likelihood requires solving a probability-flow ODE with a Hutchinson trace estimator (Grathwohl et al., 2019), injecting variance into precisely the signal that expectile regression must fit.

Coupling-based NFs (Dinh et al., 2017) uniquely meet the requirement. The triangular Jacobian makes $\log p_\theta(g \mid s, a)$ exactly and cheaply computable in closed form, and coupling architectures are universal diffeomorphism approximators (Teshima et al., 2020), so no structural gap remains. Figure 17 (Section G.4) empirically confirms that NFs attain the lowest estimation error against the analytic future-state density among CVAE, CRL and MC C-learning. Because accurate, properly normalized Q-values are the bottleneck for trajectory stitching under sparse rewards (Figure 5), exact log-density is the property we explicitly optimize for.

**Expectile Regression for In-Distribution Optimal Q-Value Prediction.** Given accurate $Q^\beta$ estimates from NFs, we still need a mechanism to extract near-optimal behaviors from suboptimal data. The expectile regression loss (Kostrikov et al., 2022; Wu et al., 2023; Zhuang et al., 2024) asymmetrically weights prediction errors:

$$L_\tau^2(u) = |\tau - \mathbf{1}(u < 0)| \cdot u^2, \qquad (9)$$

where $\tau \in (0.5, 1)$ controls the asymmetry. When $\tau > 0.5$, the loss penalizes underestimation more heavily, causing the learned value to concentrate on the upper portion of the empirical distribution. Applying this to Q-value prediction, we define:

$$\mathcal{L}_Q = \mathbb{E}_{(s_t, a_t, g) \sim \mathcal{D}} \left[ L_\tau^2 \left( Q_\theta^\beta(s_t, a_t, g) - \hat{Q}_\phi(s_t, g) \right) \right], \tag{10}$$

where $\hat{Q}_\phi(s_t, g)$ is the Q-value predicted by the Hybrid Attention-Mamba transformer with parameters $\phi$, and $Q_\theta^\beta(s_t, a_t, g)$ is the target from the NFs-based critic (Equation (4)). Our theoretical analysis (Section 3.5) demonstrates that

that this enables our sequential model, **QHyer**, to predict Q-values that approach the in-distribution maximum. These predictions correspond to the high-Q segments shown in Figure 2 (d), which are essential for trajectory stitching.

### 3.2. Limitation 2: Temporal Modeling Requires Content-Adaptive History Compression

Beyond the conditioning signal, effective sequence modeling for Offline GCRL demands architectures that can capture heterogeneous temporal dependencies inherent in Offline GCRL datasets.

**Why Offline GCRL Data Exhibits Variable-Length Historical Dependencies.** Offline GCRL datasets exhibit different temporal structures depending on behavior policy properties. As documented in OGBench (Park et al., 2025), the manipulation suite provides two representative dataset types: `play` datasets collected by non-Markovian expert policies with temporally correlated noise where the behavior policy follows $\beta(a_t|s_t, h_{<t})$, and `noisy` datasets collected by Markovian expert policies with uncorrelated Gaussian noise where $\beta(a_t|s_t)$ depends only on the current state. The `play` data demands extended memory for action coherence, while `noisy` data requires only short-term local information. A principled solution must adapt to both properties without manual tuning.

**Why Convolution Cannot Address Variable-Length Dependencies.** To address the inherent tension of datasets exhibiting the two aforementioned properties, both LSDT (Wang et al., 2025) and DMixer (Zheng et al., 2025) incorporate attention and convolution as parallel branches. Convolution-based local modeling computes features through causal convolution with fixed-size kernels:

$$y_t = \text{Conv1d}(\mathbf{x})_t = \sum_{j=0}^{k-1} w_j \cdot x_{t-j}, \tag{11}$$

where $k$ is the fixed kernel size and $w_j$ are input-independent weights. When convolution serves as the *final output* of a branch, this creates two fundamental limitations. First, convolution imposes a fixed receptive field set by the chosen kernel (and any dilation/stacking), making the effective context length a hand-tuned architectural prior that is sensitive to hyperparameters and often fails to transfer across datasets with different temporal dependencies. Second, even when convolution layers are stacked, each layer's receptive field is fixed at design time and therefore the resulting effective memory cannot adapt to varying dependency lengths within or across datasets. In particular, on non-Markovian trajectories where the relevant cues lie beyond this window, the local branch becomes weakly informative and is often down-weighted by fusion.

**Why Mamba Enables Content-Adaptive History Com-**

**pression.** To address the fixed-window bottleneck of a convolutional short-term branch, we adopt a Mamba-style selective SSM module, following DMamba (Ota, 2024). A DMamba block combines (i) a lightweight causal convolution that mixes nearby tokens and produces local features $x'_t$ (and gating signals), with (ii) a selective state-space update (Equations (7) and (8)) that propagates a recurrent state across the entire prefix. Importantly, the effective memory is not determined by the convolutional kernel, but by the input-dependent SSM dynamics (via the selective discretization), which enables smooth, learned forgetting/retention over history. As a result, compared to using convolution as the branch output, DMamba provides a content-adaptive mechanism to compress long-range context into a compact state, reducing sensitivity to hand-tuned receptive fields and improving robustness on non-Markovian segments where disambiguating cues may lie beyond any fixed local window.

To make the adaptive history modeling explicit, we expand the SSM recurrence (Equation (7)) to express the output at timestep $t$:

$$y_t = \sum_{i=0}^{t} C_t \left( \prod_{j=i+1}^{t} \bar{A}_j \right) \bar{B}_i x'_i, \tag{12}$$

where $x'_i$ is the convolution-extracted feature at step $i$. The influence of historical input $x'_i$ on current output $y_t$ is governed by the cumulative decay $\prod_{j=i+1}^{t} \bar{A}_j$. Critically, through the selective mechanism (Equation (8)), the discretization step $\Delta$ is input-dependent:

$$\bar{A}_t = \exp(\Delta_t \cdot A), \quad \text{where} \quad \Delta_t = \text{softplus}(\text{Linear}_\Delta(x'_t)), \tag{13}$$

and $A < 0$ is a negative real-valued (or diagonal-negative)

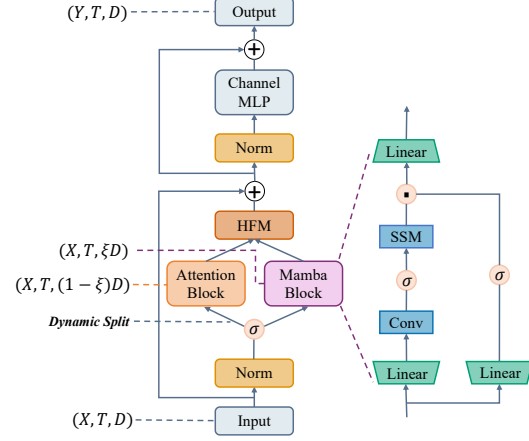

**Figure 3.** Hybrid Attention-Mamba Block.

state transition parameter, following Gu & Dao (2024). This creates **content-adaptive** effective memory: when $x'_t$ yields small $\Delta_t$, the decay $\bar{A}_t = \exp(\Delta_t \cdot A) \approx 1$ preserves long-range history suitable for `play` data; when $x'_t$ yields large $\Delta_t$, $\bar{A}_t \approx 0$ retains only local context appropriate

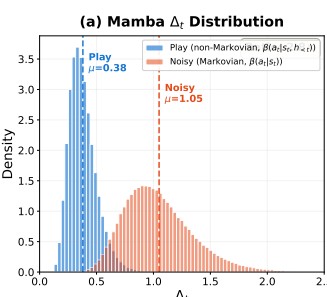 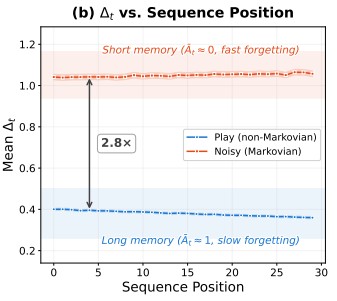 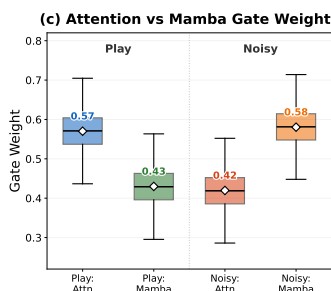

*Figure 4.* **Content-adaptive $\Delta_t$ on `cube-single`. Left:** $\Delta_t$ distribution. **Center:** mean $\Delta_t$ across sequence positions. **Right:** learned attention/Mamba gate weights. Statistics over 50 batches of batch size 256.

for `noisy` data. In contrast, convolution imposes a fixed influence $w_j$ for $j < k$ and zero beyond, resulting in hard, input-independent truncation. The key distinction is that Mamba provides smooth, learned decay, while convolution enforces hard, fixed truncation.

**Hybrid Architecture with Attention-Mamba.** As illustrated in Figure 3, we design a Hybrid Attention-Mamba architecture with two parallel branches: attention for global goal-directed planning and Mamba for temporal dynamics modeling. The outputs are fused through a learnable gating mechanism that computes a scalar weight $\alpha = \sigma(\mathbf{w}^T\mathbf{x} + b)$ to combine branch outputs: $\mathbf{y} = \alpha \cdot \mathbf{y}_{\text{attn}} + (1-\alpha) \cdot \mathbf{y}_{\text{Mamba}}$. This enables complementary specialization across both `play` and `noisy` datasets.

Figure 4 visualizes this adaptation on `cube-single`. On `play`, smaller $\Delta_t$ preserves an effective memory of about 12 steps and the gate favors attention. On `noisy`, larger $\Delta_t$ contracts memory to about 3 steps and the gate favors Mamba. The underlying reason is that $\Delta_t = \text{softplus}(\text{Linear}_\Delta(x'_t))$ is input-dependent, so the effective memory adapts to the local temporal correlation of the data, which a convolution's input-independent receptive field cannot achieve.

### 3.3. Concatenated State-Goal Tokenization Strategy

We represent each state-goal pair as a concatenated token $[s_t; g]$ rather than separate tokens. Combined with NFs-based Q-value conditioning, the input sequence becomes: $\tau = (Q_1, [s_1; g], a_1, Q_2, [s_2; g], a_2, \ldots, Q_T, [s_T; g], a_T)$, where $Q_t = \log p_\theta(g|s_t, a_t)$ is the NFs-estimated Q-value. This design ensures that goal information is directly available at each decision point without increasing the sequence length from $3T$ to $4T$, thereby avoiding the quadratic computational overhead in attention. Detailed visual explanation of this tokenization strategy is provided in Section G.1.

### 3.4. Training and Inference

**Training.** We train **QHyer** end-to-end with three losses:

$$\mathcal{L}_{\text{QHyer}} = \lambda_{\text{critic}}\mathcal{L}_{\text{NFs}} + \lambda_{\text{BC}}\mathcal{L}_{\text{BC}} + \lambda_Q\mathcal{L}_Q, \quad (14)$$

where $\mathcal{L}_{\text{BC}}$ is the behavior cloning loss that predicts actions conditioned on Q-values instead of RTG:

$$\mathcal{L}_{\text{BC}} = -\mathbb{E}_{(s_t,a_t,g)\sim\mathcal{D}}\left[\log \pi_\theta(a_t|Q_t, [s_t; g])\right]. \quad (15)$$

**Inference.** QHyer performs two-stage autoregressive generation: (1) predict maximum Q-value $\hat{Q}(s_t, g)$ from current context; (2) predict optimal action conditioned on the predicted maximum Q-value. The detailed algorithm is provided in Appendix D.

### 3.5. Theoretical Analysis

We establish convergence guarantees for **QHyer**: expectile regression yields near-optimal Q-values, and the learned policy achieves a bounded suboptimality gap relative to the in-distribution optimal stitched policy, with explicit dependence on the sample size, NFs accuracy, and dataset coverage.

**Setup.** Let $Q^\beta(\tau, g, h)$ denote the goal-reaching probability conditioned on history. The in-distribution optimal Q-value $Q^\star(s, a, g, h) := \max_{\tau \in \mathcal{T}^\beta:(s_h,a_h)=(s,a)} Q^\beta(\tau, g, h)$ represents the maximum achievable within the behavior policy's support. We assume: (i) Q-value coverage with constant $\tilde{c} \in (0, 1]$, measuring the minimum density ratio of optimal actions in the dataset; (ii) bounded NFs error $\epsilon_{\text{NFs}}$; (iii) bounded function class with approximation error $\delta_{\text{approx}}$. Full definitions are in Appendix B.

**Theorem 3.1** (Convergence of Expectile Regression to In-Distribution Optimal Q-Value)**.** *Under Q-value coverage with constant $\tilde{c} \in (0, 1]$ and sample size satisfying Equation (37) in the Appendix, for $\tau \in (0.5, 1)$, the expectile estimator satisfies $|Q^\star - \hat{Q}^\tau| \leq \epsilon_\tau$ with high probability, where the bias term $\epsilon_\tau := \frac{(1-\tau)(Q^\star - Q_{\min})}{\tau \cdot \tilde{c}/2 + (1-\tau)(1-\tilde{c}/2)}$ decreases as $\tau$ increases, at the cost of requiring more samples for variance control.*

**Theorem 3.2** (Convergence to In-Distribution Optimal Stitched Policy)**.** *Under assumptions (i) to (iii), the learned policy $\hat{\pi}^\star_\mathcal{D}$ satisfies:*

$$J(\pi^\star_\beta) - J(\hat{\pi}^\star_\mathcal{D}) \leq \underbrace{\mathcal{O}(N^{-1/4})}_{\text{policy}} + \underbrace{\sqrt{\delta_{approx}}}_{\text{MLE}} + \underbrace{\mathcal{O}(\sqrt{\epsilon_{NFs}} + \epsilon_\tau)}_{\text{Q-value}}.$$

$$(16)$$

Complete proofs are in Appendix C.

# 4. Experiments

We extensively evaluate **QHyer**'s effectiveness and conduct ablation studies on both non-Markovian and Markovian offline GCRL datasets.

**Datasets.** We consider two widely used benchmarks. For OGBench (Park et al., 2025), we evaluate on manipulation tasks including `cube`, `scene`, and `puzzle` environments with both `play` (non-Markovian) and `noisy` (Markovian) datasets. For D4RL (Fu et al., 2020), we evaluate on `Maze` (non-Markovian) tasks. A detailed introduction to these environments is presented in Appendix F.

**Baselines.** We compare **QHyer** against three categories of methods: (1) sequence modeling methods including DT (Chen et al., 2021), EDT (Wu et al., 2023), GDT (Hu et al., 2023), QDT (Yamagata et al., 2023), CGDT (Wang et al., 2024), Reinformer (Zhuang et al., 2024), DC (Kim et al., 2024b), DMamba (Ota, 2024), QT (Hu et al., 2024), LSDT (Wang et al., 2025), DMixer (Zheng et al., 2025), and VDT (Zheng et al., 2026); (2) TD-based methods including CQL (Kumar et al., 2020) and IQL (Kostrikov et al., 2022); (3) offline GCRL methods including GCBC (Ghosh et al., 2021), GCIVL, GCIQL (Kostrikov et al., 2022), QRL (Wang et al., 2023), CRL (Eysenbach et al., 2022), HIQL (Park et al., 2023), SAW (Zhou & Kao, 2026), OTA (Ahn et al., 2026), and Eik-HiQRL (Giammarino & Qureshi, 2026). For completeness, Section G.6 additionally reports comparisons against four recent offline RL methods (i.e., QCFQL (Li et al., 2026), SHARSA (Park et al., 2026a), Transitive RL (Park et al., 2026b), DEAS (Kim et al., 2026), ) adapted to GCRL with HER, as well as GAS (Baek et al., 2025) on navigation and manipulation tasks.

## 4.1. OGBench Results

Table 1 validates our core claims about sequence modeling for non-Markovian Offline GCRL. On `play` datasets collected by non-Markovian expert policies, **QHyer** substantially outperforms all baselines across manipulation tasks. Hierarchical methods (HIQL, SAW, OTA) underperform on state-based `play` datasets because their subgoal decomposition assumes Markovian transitions between subgoals, an assumption violated when behavior policies exhibit temporal correlations. Eik-HiQRL further suffers from exponential quasimetric approximation error in high-dimensional spaces (Giammarino & Qureshi, 2026), limiting its effectiveness across both state-based and visual manipulation tasks. TD-based hierarchical methods (HIQL, SAW, OTA) achieve competitive performance on `visual` tasks because hierarchical value functions provide representation learning signals beneficial for pixel inputs. On `noisy` datasets,

**QHyer** maintains competitive performance through adaptive gating between attention and Mamba branches.

## 4.2. D4RL Results

Table 2 confirms **QHyer**'s advantages on long-horizon navigation tasks where trajectory stitching is essential. **QHyer** consistently outperforms both TD-based methods and sequence modeling baselines, with the most pronounced gains on large mazes requiring extensive stitching. Vanilla DT and its variants (EDT, DC) achieve near-zero performance on medium and large mazes, directly confirming our analysis in Section 3.1. RTG under sparse goal-conditioned rewards reduces to a near-binary signal that provides no discriminative information for stitching trajectories. Conversely, the strong performance of **QHyer** in this setting also demonstrates the effectiveness of our method on non-Markovian locomotion tasks.

## 4.3. Ablation Studies

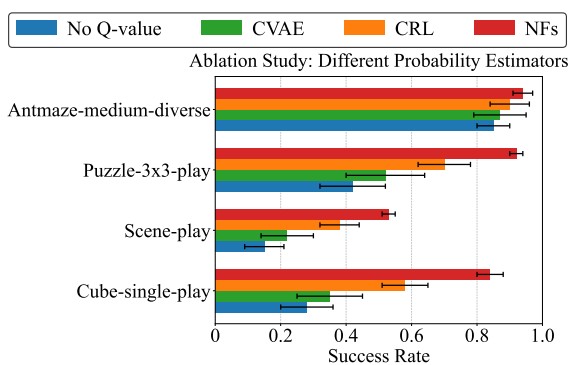

*Figure 5.* Ablation Study on Various Q-value Estimators and the Impact of Not Estimating Q-value on **QHyer**.

**Q: How does the Q-value estimator affect performance?**

**A:** Figure 5 shows a consistent ordering: No Q < CVAE < CRL < NFs, which directly reflects the relationship between density estimation accuracy and policy quality established in Section 3.1. Without Q-values, the model degenerates to behavior cloning that cannot distinguish states by their proximity to goals under sparse rewards. CVAE introduces systematic bias through the ELBO gap, distorting the goal-reaching probability landscape. CRL improves through contrastive objectives but inherits negative sampling bias that underestimates probabilities for distant goals. NFs achieve the best performance by computing exact likelihoods through invertible transformations (Equation (2)), enabling accurate identification of high-value state-action pairs via expectile regression. This mechanism is essential for extracting optimal behaviors from suboptimal data.

**Q: Does the architecture alone improve performance?**

**A:** Figure 6 isolates the architectural contribution by removing NFs-based Q-conditioning from all methods, using

*Table 1.* **Results on OGBench manipulation tasks.** Average success rate (%) across 5 test-time goals. Results averaged over 8 seeds (4 for pixel-based). Orange = best, underline = second best.

| Env | Type | Dataset | Hierarchical Policy | | | | Flat Policy | | | | | |
|---|---|---|---|---|---|---|---|---|---|---|---|---|
| | | | HIQL | SAW | OTA | Eik-HiQRL | GCBC | GCIVL | GCIQL | QRL | CRL | QHyer |
| cube | play | single | 15 ±3 | 23 ±2 | 13 ±1 | 0 ±0 | 6 ±2 | 53 ±4 | 68 ±6 | 5 ±1 | 19 ±2 | **84** ±4 |
| | | double | 6 ±2 | 26 ±3 | 2 ±1 | 0 ±0 | 1 ±1 | 36 ±3 | 40 ±5 | 1 ±0 | 10 ±2 | **56** ±2 |
| | | triple | 3 ±1 | 19 ±4 | 1 ±0 | 0 ±0 | 1 ±1 | 1 ±0 | 3 ±1 | 0 ±0 | 4 ±1 | **10** ±5 |
| | | quadruple | 0 ±0 | 0 ±0 | 0 ±0 | 0 ±0 | 0 ±0 | 0 ±0 | 0 ±0 | 0 ±0 | 2 ±1 | **2** ±1 |
| | | *Total* | *24* | *68* | *16* | *0* | *8* | *90* | *111* | *6* | *35* | *152* |
| | noisy | single | 41 ±6 | 38 ±2 | 40 ±2 | 2 ±1 | 8 ±3 | 71 ±9 | **99** ±1 | 25 ±6 | 38 ±2 | 95 ±5 |
| | | double | 2 ±1 | 12 ±1 | 5 ±2 | 0 ±0 | 1 ±1 | 14 ±3 | 23 ±3 | 3 ±1 | 2 ±1 | **30** ±4 |
| | | triple | 2 ±1 | 13 ±1 | 1 ±0 | 0 ±0 | 1 ±1 | 9 ±1 | 2 ±1 | 1 ±0 | 3 ±1 | **14** ±1 |
| | | quadruple | 0 ±0 | 1 ±0 | 0 ±0 | 0 ±0 | 0 ±0 | 0 ±0 | 0 ±0 | 0 ±0 | 0 ±0 | **6** ±4 |
| | | *Total* | *45* | *64* | *46* | *2* | *10* | *94* | *124* | *29* | *43* | *145* |
| scene | play | scene | 38 ±3 | **58** ±3 | 19 ±4 | 11 ±3 | 5 ±1 | 42 ±4 | 51 ±4 | 5 ±1 | 19 ±2 | 53 ±2 |
| | noisy | scene | **25** ±4 | 25 ±2 | 12 ±1 | 12 ±0 | 1 ±1 | 26 ±5 | 26 ±2 | 9 ±2 | 1 ±1 | 25 ±5 |
| puzzle | play | 3x3 | 12 ±2 | 6 ±1 | 21 ±5 | 9 ±0 | 2 ±0 | 6 ±1 | **95** ±1 | 1 ±0 | 3 ±1 | 92 ±2 |
| | | 4x4 | 7 ±2 | 6 ±1 | 5 ±2 | 2 ±1 | 0 ±0 | 13 ±2 | 26 ±3 | 0 ±0 | 0 ±0 | **28** ±5 |
| | | 4x5 | 4 ±1 | 2 ±1 | 2 ±1 | 0 ±0 | 0 ±0 | 7 ±1 | 14 ±1 | 0 ±0 | 1 ±0 | **31** ±1 |
| | | 4x6 | 3 ±1 | 5 ±0 | 1 ±1 | 0 ±0 | 0 ±0 | 10 ±2 | 12 ±1 | 0 ±0 | 4 ±1 | **18** ±2 |
| | | *Total* | *26* | *19* | *29* | *11* | *2* | *36* | *147* | *1* | *8* | *169* |
| | noisy | 3x3 | 51 ±11 | 78 ±39 | 53 ±6 | 6 ±2 | 1 ±0 | 42 ±19 | **94** ±3 | 0 ±0 | 30 ±6 | 89 ±8 |
| | | 4x4 | 16 ±4 | 0 ±0 | 0 ±0 | 3 ±1 | 0 ±0 | 20 ±3 | 29 ±7 | 0 ±0 | 0 ±0 | **33** ±6 |
| | | 4x5 | 5 ±1 | 15 ±1 | 1 ±1 | 2 ±2 | 0 ±0 | 19 ±0 | 19 ±0 | 0 ±0 | 3 ±2 | **26** ±2 |
| | | 4x6 | 2 ±1 | 11 ±2 | 0 ±0 | 0 ±0 | 0 ±0 | 17 ±2 | 18 ±2 | 0 ±0 | 6 ±3 | **24** ±3 |
| | | *Total* | *74* | *104* | *54* | *11* | *1* | *98* | *160* | *0* | *39* | *172* |
| visual-cube | play | single | **89** ±0 | 88 ±3 | 40 ±4 | / | 5 ±1 | 60 ±5 | 30 ±5 | 41 ±15 | 31 ±15 | 42 ±2 |
| | | double | 39 ±2 | **40** ±3 | 36 ±17 | / | 1 ±1 | 10 ±2 | 1 ±1 | 5 ±0 | 2 ±1 | 37 ±3 |
| | | triple | **21** ±0 | 20 ±1 | 24 ±2 | / | 15 ±2 | 14 ±2 | 15 ±1 | 16 ±1 | 17 ±2 | 20 ±4 |
| | | *Total* | *149* | *148* | *100* | */* | *21* | *84* | *46* | *62* | *50* | *99* |
| visual-scene | play | scene | 49 ±4 | 47 ±6 | 62 ±3 | / | 12 ±2 | 25 ±3 | 12 ±2 | 10 ±1 | 11 ±2 | **96** ±1 |
| | noisy | scene | 50 ±1 | 54 ±3 | **63** ±2 | / | 13 ±2 | 23 ±2 | 12 ±4 | 2 ±0 | 15 ±2 | 36 ±4 |

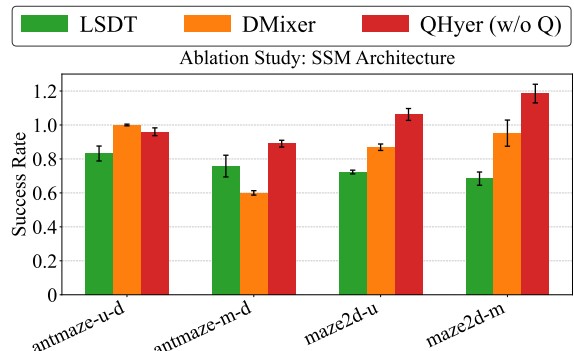

*Figure 6.* Ablation study on SSM variants for temporal modeling.'-u','-m' and '-d' denote umaze, medium, and diverse, respectively.

content, rather than relying on predefined kernel sizes or discrete selection thresholds. Combined with the results in Figure 5, this demonstrates that **QHyer**'s two innovations provide complementary and additive performance improvements.

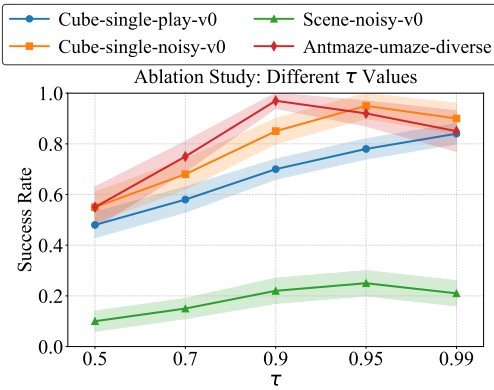

*Figure 7.* Ablation study on the expectile parameter $\tau$ for Q-value prediction.

**Q: How should the expectile parameter $\tau$ be selected?**

**A:** Figure 7 shows monotonic improvement from $\tau = 0.5$ to $\tau = 0.9$, with optimal performance at $\tau \in [0.9, 0.95]$. This validates Theorem 3.1: higher $\tau$ reduces the bias term $\epsilon_\tau$ by focusing on upper expectiles, enabling identification of high-Q segments for trajectory stitching that RTG's trajectory-

standard RTG instead. The results reveal a consistent ordering: LSDT < DMixer < **QHyer** across both AntMaze and Maze2d environments. This validates that the performance gains stem from both innovations independently. LSDT's Dynamic Convolution branch is limited by its fixed kernel size, which cannot adaptively capture dependencies of varying ranges. DMixer's token-level selection mechanism improves upon LSDT but can disrupt the continuity of action patterns through discrete token dropping. In contrast, **QHyer**'s Mamba branch maintains compressed hidden states that enable content-adaptive dependency modeling. The selective SSM parameters (B, C, $\Delta$) dynamically determine how much historical context to retain based on input

*Table 2.* **Results on D4RL.** Normalized scores (5 seeds) from original papers except QHyer. Orange = best, underline = second best.

| Antmaze-v2 | RL | | Supervised Learning | | | | | | | | | |
|---|---|---|---|---|---|---|---|---|---|---|---|---|
| | CQL | IQL | DT | RvS | EDT | CGDT | DC | DMamba | Reinformer | QT | LSDT | QHyer |
| umaze | 74.0 | 87.5 | 64.5 | 65.4 | 67.8 | 71.0 | 85.0 | 81.8 | 84.4 | 96.7 | 80.0 | **98.4**±1.9 |
| umaze-diverse | 84.0 | 62.2 | 60.5 | 60.9 | 58.3 | 71.0 | 78.5 | 71.6 | 65.8 | 96.7 | 83.2 | **97.1**±2.3 |
| medium-play | 61.2 | 71.2 | 0.8 | 58.1 | 0.0 | / | 1.5 | 79.6 | 13.2 | / | 85.5 | **92.2**±3.5 |
| medium-diverse | 53.7 | 70.0 | 0.5 | 67.3 | 0.0 | / | 0.0 | 83.2 | 10.6 | 59.3 | 75.8 | **94.0**±2.7 |
| large-play | 15.8 | 39.6 | 0.0 | 32.4 | 0.0 | / | 0.0 | 23.2 | 0.4 | / | 0.0 | **44.2**±1.9 |
| large-diverse | 14.9 | 47.5 | 0.0 | 32.9 | 0.0 | / | 0.0 | 34.6 | 0.4 | 53.3 | 0.0 | **57.5**±13.5 |
| *Total* | 303.6 | 378.0 | 126.3 | 317.0 | 126.1 | / | 165.0 | 374.0 | 174.8 | / | 324.5 | *483.4* |
| Maze2d | CQL | IQL | DT | QDT | GDT | VDT | DC | DMamba | DMixer | QT | LSDT | QHyer |
| umaze | 94.7 | 74.0 | 31.0 | 57.3 | 50.4 | 60.3 | 20.1 | 83.4 | 86.9 | 105.4 | 72.3 | **118.5**±1.9 |
| medium | 41.8 | 84.0 | 8.2 | 13.3 | 7.8 | 88.0 | 38.2 | 98.7 | 95.2 | 172.0 | 68.4 | **173.0**±11.9 |
| *Total* | 136.5 | 158.0 | 39.2 | 70.6 | 58.2 | 148.3 | 58.3 | 182.1 | 182.1 | 277.4 | 140.7 | *291.5* |

dependence fundamentally cannot provide. However, extreme $\tau$ causes degradation by over-concentrating on too few samples, increasing estimation variance. This aligns with our theoretical analysis where $\epsilon_\tau$ depends on both $\tau$ and coverage $\tilde{c}$. As $\tau$ approaches 1, sensitivity to coverage limitations amplifies. We use $\tau=0.9$ for low-coverage (play) and $\tau=0.95$ for high-coverage (noisy) data.

**Q: Is the Hybrid architecture's gain actually architectural, or is it confounded by Q-conditioning, and does Mamba truly adapt its memory to data type?**

**A:** We answer both jointly. Table 3 fixes NFs Q-conditioning and varies *only* the backbone. On the non-Markovian cube-single-play dataset, the Attention-only, Mamba-only, and Hybrid backbones achieve success rates of 74%, 80%, and 84%, respectively. On the Markovian cube-single-noisy dataset, the corresponding success rates are 60%, 91%, and 95%. The Hybrid backbone outperforms the best single-branch variant by 4 to 5 percentage points under *both* data regimes, indicating that the scalar gate captures genuinely complementary specialization rather than interpolating between two near-identical branches.

*Table 3.* **Backbone ablation with identical NFs Q-conditioning.** Mean success rate (%) over 4 seeds. Orange = best, underline = second best.

| Environment | Attention-only | Mamba-only | Hybrid (QHyer) |
|---|---|---|---|
| cube-single-play (non-Markov.) | 74 ±1 | 80 ±2 | **84** ±4 |
| cube-single-noisy (Markov.) | 60 ±3 | 91 ±3 | **95** ±5 |

Table 4 then explains *why*, by extracting Mamba's $\Delta_t$ and the learned gate weight from the trained model (cf. Figure 4). On play, mean $\Delta_t=0.38$ and $\bar{A}_t=0.92$, the SSM retains about 12 steps of effective history, and the gate shifts 0.57 of capacity to attention for global goal-directed reasoning. On noisy, $\Delta_t=1.05$ and $\bar{A}_t=0.61$, memory collapses to about 3 steps, and the gate shifts 0.58 to Mamba. The essential reason is that Mamba's selective mechanism makes $\Delta_t$ a function of the input, so effective memory varies per-token with the local temporal correlation, which

convolution-based hybrids (LSDT, DMixer) cannot produce because their receptive field is an architectural constant.

*Table 4.* $\Delta_t$ **and gate statistics on cube-single**, extracted from trained **QHyer** (50 batches, batch 256). Effective memory length is the largest $k$ with $\prod_{j=1}^{k} \bar{A}_{t-j+1} > 0.5$.

| Metric | play (non-Markov.) | noisy (Markov.) |
|---|---|---|
| Mean $\Delta_t$ | 0.38 | 1.05 |
| Std $\Delta_t$ | 0.12 | 0.31 |
| Mean $\bar{A}_t = \exp(\Delta_t \cdot A)$ | 0.92 | 0.61 |
| Effective memory (steps) | $\sim 12$ | $\sim 3$ |
| Gate weight (Attention) | 0.57 | 0.42 |
| Gate weight (Mamba) | 0.43 | 0.58 |

Combined with Figure 5 (NFs vs. CVAE/CRL/No-Q), Figure 6 (backbone with RTG), Table 3 (backbone with NFs), and Table 4 (mechanism), these ablations together establish that each of **QHyer**'s two innovations is necessary and that the two components are genuinely complementary.

## 5. Conclusion

We propose **QHyer**, the first sequence modeling framework for non-Markovian offline GCRL that addresses two fundamental limitations: replacing *trajectory-dependent* RTG with *state-action-dependent* Q-values estimated via NFs for effective trajectory stitching, and introducing a Hybrid Attention-Mamba architecture for content-adaptive temporal modeling. Experiments on OGBench and D4RL demonstrate state-of-the-art performance, particularly on non-Markovian datasets.

**Limitations and future work. QHyer** remains constrained on visual-noisy, where Markovian behavior neutralizes the non-Markovian modeling advantage and pixel-level NFs density estimation becomes the dominant source of error. Promising future directions include robust visual density estimation and extension of the deterministic-transition theory (Appendix B) to stochastic environments.

## Acknowledgments

This work was supported in part by the Brain Science and Brain-like Intelligence Technology — National Science and Technology Major Project (Grant No. 2022ZD0208800) and the Provincial Key Research and Development Program (Project No. 2023-YBGY-033).

## Impact Statement

This paper presents work whose goal is to advance the field of Machine Learning. There are many potential societal consequences of our work, none which we feel must be specifically highlighted here.

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

# A. Related Work

**Offline Goal-Conditioned RL (GCRL).** Offline GCRL aims to learn goal-reaching policies from static datasets without environment interaction. Existing approaches can be categorized into several paradigms: goal-conditioned hindsight relabeling (Andrychowicz et al., 2017), hierarchical or subgoal-based learning (Park et al., 2023; Ahn et al., 2026; Giammarino & Qureshi, 2026; Zhou & Kao, 2026; Lei et al., 2025), graph-based planning (Yoon et al., 2024; Eysenbach et al., 2024), metric learning (Wang et al., 2023; Park et al., 2024; Myers et al., 2024; 2026), dual optimization (Ma et al.; Sikchi et al., 2024), generative modeling (Hong et al., 2023; Reuss et al., 2023; Jain & Ravanbakhsh, 2024), and test-time adaption (Opryshko et al., 2025). However, these methods predominantly assume that the offline data follows Markovian properties—that the optimal action depends solely on the current state and goal. Existing methods struggle on such non-Markovian datasets because they cannot capture the temporal dependencies that govern the behavior policy's decisions. In contrast, **QHyer** explicitly models these dependencies through sequence modeling framework.

**NFs in RL.** NFs are invertible generative models that enable exact likelihood computation and efficient sampling (Dinh et al., 2014; 2017; Kingma & Dhariwal, 2018). Recent work has demonstrated their effectiveness in RL for policy modeling (Singh et al., 2021; Ward et al., 2019; Chao et al., 2024) and Q-function estimation. Chao et al. (2024) propose Energy-Based Normalizing Flows (EBFlow) that unify policy evaluation and improvement into a single objective for maximum entropy RL, enabling exact soft value function calculation without Monte Carlo approximation. Brahmanage et al. (2023) leverage NFs to learn invertible mappings between feasible action spaces and Gaussian latent spaces for action-constrained policy gradient methods.

For offline settings, Akimov et al. (2022) use NFs-based action encoders to construct conservative action spaces, addressing distributional shift without explicit regularization. Notably, Ghugare & Eysenbach (2026) show that NFs can serve as Q-functions in GCRL by modeling the discounted state occupancy distribution, achieving strong performance on offline GCRL benchmarks with a simple feedforward architecture. However, their approach cannot capture temporal dependencies in non-Markovian datasets. Our work integrates NFs-based Q-estimation into a sequence modeling framework, enabling both accurate value estimation and temporal dependency modeling.

**Sequence Modeling in Offline RL.** Decision Transformer (DT) (Chen et al., 2021) reformulates offline RL as conditional sequence modeling, where actions are generated conditioned on desired returns and past states. This paradigm has spurred extensive research, which can be broadly categorized into two directions: *value-enhanced methods* and *architectural innovations*.

*Value-enhanced methods* integrate reinforcement learning principles to address DT's fundamental limitation in stitching sub-optimal trajectories (Brandfonbrener et al., 2022). For instance, Q-learning Decision Transformer (QDT) (Yamagata et al., 2023) employs dynamic programming for optimal path synthesis. Critic-Guided Decision Transformer (CGDT) (Wang et al., 2024) incorporates a value-based critic to align expected returns with target returns. Q-value Regularized Transformer (QT) (Hu et al., 2024) introduces explicit Q-value regularization to tackle long-horizon and sparse-reward tasks. Reinformer (Zhuang et al., 2024) utilizes expectile regression for maximizing returns, while Value-Guided Decision Transformer (VDT) (Zheng et al., 2026) leverages value functions for advantage-weighted behavior regularization. These methods primarily employ TD-learning for Q-value estimation and use value functions as auxiliary losses or regularizers. In contrast, QHyer estimates Q-values via NFs with Monte Carlo learning and directly uses them as conditioning tokens to replace RTG.

*Architectural innovations* aim to more effectively capture the heterogeneous temporal patterns present in offline datasets. Elastic Decision Transformer (EDT) (Wu et al., 2023) enables adaptive history length selection to facilitate stitching. Graph Decision Transformer (GDT) (Hu et al., 2023) structures input sequences as causal graphs with relation-enhanced attention mechanisms. Decision Convformer (DC) (Kim et al., 2024b) replaces attention with causal convolution filters to model local, Markovian associations efficiently. Decision Mamba (DMamba) (Ota, 2024) substitutes attention with selective state space models for linear-time sequence modeling. Long-Short Decision Transformer (LSDT) (Wang et al., 2025) combines attention with dynamic convolution using a fixed capacity ratio, and Decision Mixer (DMixer) (Zheng et al., 2025) integrates long-term and local features via dynamic token selection. QHyer introduces a Hybrid Attention-Mamba architecture with learnable gating that dynamically allocates capacity, allowing attention to handle global goal-directed planning while Mamba captures local temporal patterns with content-adaptive memory.

Our work deviates from both value-based and architectural innovations methods. To our knowledge, this is the first work to unlock the potential of sequence modeling for Offline GCRL.

# B. Notation and Assumptions

## B.1. Notation

We consider goal-conditioned episodic MDPs with finite horizon $H$. **Following Reinforced Return-conditioned Supervised Learning ($\mathbf{R}^2\mathbf{CSL}$) (Liu et al., 2025), we assume deterministic transitions**, i.e., given state $s$ and action $a$, the next state $s' = P(s, a)$ is uniquely determined. This ensures that the in-distribution optimal Q-value $Q^\star(s, a, g, h)$ is well-defined as a unique value. Extension to stochastic environments is an important direction for future work.

| Symbol | Definition |
|---|---|
| *Spaces and Indices* | |
| $\mathcal{S}, \mathcal{A}, \mathcal{G}$ | State space, action space, goal space |
| $H$ | Episode horizon (total number of stages per episode) |
| $h \in [H] := \{1, \ldots, H\}$ | Stage index (timestep within an episode) |
| $\phi : \mathcal{S} \to \mathcal{G}$ | Goal mapping; in our experiments, $\mathcal{G} \subseteq \mathcal{S}$ |
| *Policies and Distributions* | |
| $\beta$ | Behavior policy that generated the offline dataset |
| $\pi_\beta^\star$ | In-distribution optimal stitched policy (Eq. 19) |
| $\hat{\pi}_\mathcal{D}^\star$ | Learned policy from QHyer |
| $d_h^\beta(s)$ | State visitation probability at stage $h$ under policy $\beta$ |
| $d_{\min}^\beta$ | Minimum positive state visitation: $\min_{h,s}\{d_h^\beta(s) \mid d_h^\beta(s) > 0\}$ |
| $c_\beta^\star$ | Distribution mismatch coefficient (Assumption B.5) |
| *Q-Values (Key Distinction)* | |
| $Q^\beta(s, a, g)$ | True goal-reaching probability under $\beta$: $p_+^\beta(g \mid s, a)$ |
| $Q^\star(s, a, g, h)$ | In-distribution optimal Q-value: $\max_{\tau:(s_h,a_h)=(s,a)} Q^\beta(\tau, g, h)$ |
| $\hat{Q}_\theta^\beta(s, a, g)$ | NFs estimate of $Q^\beta$, trained via Equation (5) |
| $\hat{Q}^\tau(s, a, g, h)$ | Expectile regression output on NFs-estimated Q-values |
| $\hat{Q}_\phi(s, g, h)$ | Transformer-predicted Q-value for conditioning (Equation (10)) |
| *Error Terms* | |
| $\epsilon_{\text{NFs}}$ | NFs estimation MSE (Assumption B.7) |
| $\epsilon_\tau$ | Expectile regression bias (Theorem 3.1) |
| $\delta_{\text{approx}}$ | MLE approximation error (Assumption B.3) |
| $\tilde{c}$ | Q-value coverage constant (Assumption B.4) |

**Q-Value Definition.** For a trajectory $\tau \in \mathcal{T}^\beta$ passing through $(s_h, a_h) = (s, a)$ at stage $h$, the goal-reaching Q-value is:

$$Q^\beta(\tau, g, h) := p_+^\beta(g \mid s_h, a_h) = (1 - \gamma) \sum_{t=0}^\infty \gamma^t \cdot \mathbf{1}[\phi(s_{h+t+1}) = g], \tag{17}$$

where in deterministic environments, this reduces to the discounted indicator of whether the trajectory reaches goal $g$.

**In-Distribution Optimal Q-Value:**

$$Q^\star(s, a, g, h) := \max_{\tau \in \mathcal{T}^\beta:(s_h,a_h)=(s,a)} Q^\beta(\tau, g, h). \tag{18}$$

**Optimal Stitched Policy:**

$$\pi_\beta^\star(a \mid s, g, h) := P_\beta(a \mid s, g, h, Q^\star(s, a, g, h)). \tag{19}$$

**Performance Metric:**

$$J(\pi) := \mathbb{E}_{s_1 \sim \rho, g \sim p(g)}[V_1^\pi(s_1, g)], \tag{20}$$

where $V_h^\pi(s, g) := \mathbb{E}_\pi[\sum_{t=h}^{H} r(s_t, a_t, g) \mid s_h = s]$ and $r(s, a, g) = \mathbb{1}[\phi(s') = g]$.

## B.2. Assumptions

**Assumption B.1** (Deterministic Environment). *The transition dynamics $P : \mathcal{S} \times \mathcal{A} \to \mathcal{S}$ is deterministic, i.e., given $(s, a)$, the next state $s' = P(s, a)$ is unique. This is standard in goal-conditioned RL theory (Park et al., 2025) and holds approximately in robotic manipulation tasks.*

*Remark* B.2 (Scope of Assumption B.1). Assumption B.1 constrains the *transition dynamics* $P(s' \mid s, a)$, not the behavior policy $\beta$. This is compatible with all our experimental settings. OGBench runs deterministic MuJoCo dynamics even for `noisy` datasets, where the "noise" is Gaussian perturbation of $\beta$ rather than of $P$, and D4RL mazes likewise use deterministic $P$. Non-Markovian `play` data corresponds to a history-dependent $\beta(a_t \mid s_t, h_{<t})$ over a deterministic MDP. QHyer's sequence modeling targets exactly this behavior-policy non-Markovianness, while Theorems 3.1 and 3.2 analyze stitching on the underlying MDP. The assumption matches R$^2$CSL (Liu et al., 2025) and is standard in the offline GCRL theory literature. Extension to stochastic $P$ is a genuine open problem that we flag in the conclusion.

**Assumption B.3** (Policy Class Regularity). *The policy class $\Pi$ satisfies:*

1. *$|\Pi| < \infty$ (can be relaxed to finite covering number).*

2. *For all $(a, s, g, h, Q) \in \mathcal{A} \times \mathcal{S} \times \mathcal{G} \times [H] \times [0, 1]$ and $\pi \in \Pi$: $|\log \pi(a \mid s, g, h, Q)| \leq c$.*

3. *$\min_{\pi \in \Pi} L(\pi) \leq \delta_{approx}$, where $L(\pi) := \mathbb{E}_{(s,g,Q) \sim P_\beta}[D_{KL}(P_\beta(\cdot \mid s, g, h, Q) \| \pi(\cdot \mid s, g, h, Q))]$.*

**Assumption B.4** (Q-Value Coverage). *For each $(s, a, g, h)$ in the support of $\beta$, define:*

$$\mathcal{T}_\mathcal{D}(s, a, g, h) := \{k \in [N] : (s_h^k, a_h^k) = (s, a)\}. \tag{21}$$

*For trajectory $k$, let $Q_h^k(g) := Q^\beta(\tau^k, g, h)$ be the empirical goal-reaching probability computed via hindsight relabeling. There exists $\tilde{c} \in (0, 1]$ such that:*

$$\frac{|\{k \in \mathcal{T}_\mathcal{D}(s, a, g, h) : Q_h^k(g) = Q^\star(s, a, g, h)\}|}{|\mathcal{T}_\mathcal{D}(s, a, g, h)|} \geq \tilde{c}. \tag{22}$$

*Interpretation: At least $\tilde{c}$-fraction of trajectories through $(s, a)$ achieve the optimal Q-value. Under Assumption B.1, $Q^\star$ is well-defined as the maximum over a finite set of deterministic outcomes.*

**Assumption B.5** (Distribution Mismatch). *There exists $c_\beta^\star > 0$ such that $d_h^{\star,\beta}(s)/d_h^\beta(s) \leq c_\beta^\star$ for all $(h, s) \in [H] \times \mathcal{S}_h^\beta$.*

**Assumption B.6** (Bounded Q-Values). *$Q^\beta(s, a, g) \in [0, 1]$ for all $(s, a, g)$, since it represents a probability.*

**Assumption B.7** (NFs Estimation Error). *The NFs estimator satisfies:*

$$\mathbb{E}_{(s,a,g) \sim d_h^\beta}\left[(Q^\beta(s, a, g) - \hat{Q}_\theta^\beta(s, a, g))^2\right] \leq \epsilon_{NFs}, \quad \forall h \in [H]. \tag{23}$$

**Assumption B.8** (Policy Lipschitz Continuity). *For any $(s, g, h)$, $\pi \in \Pi$, and $Q_1, Q_2 \in [0, 1]$:*

$$TV(\pi(\cdot \mid s, g, h, Q_1) \| \pi(\cdot \mid s, g, h, Q_2)) \leq L_\pi |Q_1 - Q_2|. \tag{24}$$

**Assumption B.9** (Expectile Lipschitz Stability). *Let $\mathcal{E}_\tau(\{Q_k\})$ denote the $\tau$-expectile of samples $\{Q_k\}$. For any two sample sets $\{Q_k\}$ and $\{\tilde{Q}_k\}$ with $|Q_k - \tilde{Q}_k| \leq \epsilon$ for all $k$:*

$$|\mathcal{E}_\tau(\{Q_k\}) - \mathcal{E}_\tau(\{\tilde{Q}_k\})| \leq L_Q \cdot \epsilon, \tag{25}$$

*where $L_Q \leq 1$ is a Lipschitz constant. This holds because the expectile is a weighted average of samples.*

# C. Proofs of Theoretical Results

## C.1. Proof of Theorem 3.1

*Proof.* We prove convergence of expectile regression to the in-distribution optimal Q-value, accounting for NFs estimation error.

**Problem Setup.** Fix $(s, a, g, h)$. Let $\{Q_1, \ldots, Q_K\}$ be the true Q-values across $K := |\mathcal{T}_\mathcal{D}(s, a, h)|$ trajectories. In practice, we observe NFs estimates $\{\hat{Q}_1, \ldots, \hat{Q}_K\}$ where $\hat{Q}_k = \hat{Q}_\theta^\beta(s_{h+1}^k, g)$. The expectile loss is $L_\tau^2(u) := |\tau - \mathbf{1}(u < 0)| \cdot u^2$. We define:

- $\hat{Q}^{\tau, \text{oracle}} := \arg\min_{\tilde{Q}} \sum_k L_\tau^2(Q_k - \tilde{Q})$ — expectile on **true** Q-values

- $\hat{Q}^\tau := \arg\min_{\tilde{Q}} \sum_k L_\tau^2(\hat{Q}_k - \tilde{Q})$ — expectile on **NFs-estimated** Q-values

Our goal is to bound $|Q^\star - \hat{Q}^\tau|$.

**Step 1: Decomposition via Triangle Inequality.**

$$|Q^\star - \hat{Q}^\tau| \leq \underbrace{|Q^\star - \hat{Q}^{\tau, \text{oracle}}|}_{\text{(A) Expectile bias on true Q}} + \underbrace{|\hat{Q}^{\tau, \text{oracle}} - \hat{Q}^\tau|}_{\text{(B) NFs error propagation}} . \tag{26}$$

**Step 2: Bounding Term (A) — Expectile Bias.** From the first-order condition, the expectile satisfies:

$$\hat{Q}^{\tau, \text{oracle}} = \frac{(1 - \tau)n_- \bar{Q}_- + \tau n_+ \bar{Q}_+}{(1 - \tau)n_- + \tau n_+}, \tag{27}$$

where $n_+ = |\{k : Q_k \geq \hat{Q}^{\tau, \text{oracle}}\}|$, $n_- = K - n_+$, and $\bar{Q}_\pm$ are conditional means.

*Case 1*: If $\hat{Q}^{\tau, \text{oracle}} = Q^\star$, then $|Q^\star - \hat{Q}^{\tau, \text{oracle}}| = 0 \leq \epsilon_\tau$ trivially.

*Case 2*: If $\hat{Q}^{\tau, \text{oracle}} < Q^\star$ (generic case). Since $\hat{Q}^{\tau, \text{oracle}}$ is a convex combination:

$$Q_{\min} \leq \hat{Q}^{\tau, \text{oracle}} \leq Q^\star. \tag{28}$$

By Assumption B.4, at least $\tilde{c}$-fraction achieve $Q^\star$. Using Hoeffding's inequality, with high probability, $n^\star := |\{k : Q_k = Q^\star\}| \geq \tilde{c}K/2$. Since $\hat{Q}^{\tau, \text{oracle}} < Q^\star$, all $Q^\star$-samples are in the "above" group: $n_+ \geq \tilde{c}K/2$.

Worst-case analysis with $\bar{Q}_+ = Q^\star$ and $\bar{Q}_- = Q_{\min}$:

$$Q^\star - \hat{Q}^{\tau, \text{oracle}} = \frac{(1 - \tau)n_-(Q^\star - Q_{\min})}{(1 - \tau)n_- + \tau n_+} \tag{29}$$

$$\leq \frac{(1 - \tau)(1 - \tilde{c}/2)K(Q^\star - Q_{\min})}{(1 - \tau)(1 - \tilde{c}/2)K + \tau(\tilde{c}/2)K} \tag{30}$$

$$= \frac{(1 - \tau)(Q^\star - Q_{\min})}{\tau \cdot \tilde{c}/2 + (1 - \tau)(1 - \tilde{c}/2)} =: \epsilon_\tau. \tag{31}$$

**Step 3: Bounding Term (B) — NFs Error Propagation.** By Assumption B.9, the expectile is Lipschitz in its inputs:

$$|\hat{Q}^{\tau, \text{oracle}} - \hat{Q}^\tau| = |\mathcal{E}_\tau(\{Q_k\}) - \mathcal{E}_\tau(\{\hat{Q}_k\})| \leq L_Q \cdot \max_k |Q_k - \hat{Q}_k|. \tag{32}$$

By Assumption B.7 and Markov's inequality, with high probability:

$$\max_k |Q_k - \hat{Q}_k| \leq O(\sqrt{\epsilon_{\text{NFs}}}). \tag{33}$$

Therefore, we have:

$$|\hat{Q}^{\tau, \text{oracle}} - \hat{Q}^\tau| \leq L_Q \sqrt{\epsilon_{\text{NFs}}}. \tag{34}$$

**Step 4: Combining Terms.** From Equation (26), we have:

$$|Q^\star - \hat{Q}^\tau| \le \epsilon_\tau + L_Q \sqrt{\epsilon_{\text{NFs}}}. \tag{35}$$

**Step 5: Sample Complexity.** We need uniform convergence over all $(s, a, g, h) \in \mathcal{S} \times \mathcal{A} \times \mathcal{G} \times [H]$. By union bound with $|\mathcal{G}|$ goals, we have:

*Condition 1* (sufficient visits): For each $(s, a, h)$, we need $N_h^{s,a} \ge N d_{\min}^\beta / 2$. By Hoeffding:

$$\Pr(N_h^{s,a} < N d_{\min}^\beta / 2) \le \exp(-(d_{\min}^\beta)^2 N / 2). \tag{36}$$

*Condition 2* (coverage concentration): Given $K$ visits, need $n^\star \ge \tilde{c} K / 2$.

Setting failure probability $\le \delta / (2|\mathcal{S}||\mathcal{A}||\mathcal{G}|H)$ for each tuple:

$$N \ge \max \left\{ \frac{2}{(d_{\min}^\beta)^2} \log \frac{2|\mathcal{S}||\mathcal{A}||\mathcal{G}|H}{\delta}, \frac{4}{\tilde{c}^2 d_{\min}^\beta} \log \frac{2|\mathcal{S}||\mathcal{A}||\mathcal{G}|H}{\delta} \right\}. \tag{37}$$

$\square$

### C.2. Proof of Theorem 3.2

*Proof.* We prove convergence to the optimal stitched policy with careful treatment of distribution mismatch.

**First, Performance Difference.** Since rewards are bounded in $[0, 1]$:

$$J(\pi_\beta^\star) - J(\hat{\pi}_\mathcal{D}^\star) \le H \cdot \|d^{\pi_\beta^\star} - d^{\hat{\pi}_\mathcal{D}^\star}\|_1. \tag{38}$$

**Second, Simulation Lemma.** By the simulation lemma (Kakade, 2001), we have:

$$\|d^{\pi_\beta^\star} - d^{\hat{\pi}_\mathcal{D}^\star}\|_1 \le 2 \sum_{h=1}^{H} \mathbb{E}_{s \sim d_h^{\pi_\beta^\star}} \left[ \mathrm{TV}(\pi_\beta^\star(\cdot|s) \| \hat{\pi}_\mathcal{D}^\star(\cdot|s)) \right]. \tag{39}$$

**Third, Policy Difference Decomposition.**

$$\mathrm{TV}(\pi_\beta^\star(\cdot|s, g, h) \| \hat{\pi}_\mathcal{D}^\star(\cdot|s, g, h))$$
$$\le \underbrace{\mathrm{TV}(P_\beta(\cdot|s, g, h, Q^\star) \| \hat{\pi}(\cdot|s, g, h, Q^\star))}_{\text{(I) Policy learning error}} + \underbrace{\mathrm{TV}(\hat{\pi}(\cdot|s, g, h, Q^\star) \| \hat{\pi}(\cdot|s, g, h, \hat{Q}^\tau))}_{\text{(II) Q-value error}}. \tag{40}$$

**Bounding Term (I) of Equation (40) with Correct Derivation Order.** We carefully apply the inequalities in the correct order:

*a*: Apply Jensen's inequality to the expectation of TV, we have:

$$\mathbb{E}_{s \sim d_h^{\pi_\beta^\star}}[\mathrm{TV}(P_\beta \| \hat{\pi})] \le \sqrt{\mathbb{E}_{s \sim d_h^{\pi_\beta^\star}}[\mathrm{TV}(P_\beta \| \hat{\pi})^2]}. \tag{41}$$

*b*: Apply Pinsker's inequality ($\mathrm{TV}^2 \le \frac{1}{2} D_{\mathrm{KL}}$), we have:

$$\sqrt{\mathbb{E}_{s \sim d_h^{\pi_\beta^\star}}[\mathrm{TV}^2]} \le \sqrt{\frac{1}{2} \mathbb{E}_{s \sim d_h^{\pi_\beta^\star}}[D_{\mathrm{KL}}(P_\beta \| \hat{\pi})]}. \tag{42}$$

$c$: Apply distribution mismatch (Assumption B.5), we have:

$$\mathbb{E}_{s \sim d_h^{\pi_\beta^\star}}[D_{\mathrm{KL}}] = \sum_s d_h^{\pi_\beta^\star}(s) \cdot D_{\mathrm{KL}}(P_\beta(\cdot|s)\|\hat{\pi}(\cdot|s)) \tag{43}$$

$$= \sum_s \frac{d_h^{\pi_\beta^\star}(s)}{d_h^\beta(s)} \cdot d_h^\beta(s) \cdot D_{\mathrm{KL}} \tag{44}$$

$$\leq c_\beta^\star \cdot \mathbb{E}_{s \sim d_h^\beta}[D_{\mathrm{KL}}]. \tag{45}$$

Combining a-c, we have:

$$\mathbb{E}_{s \sim d_h^{\pi_\beta^\star}}[\mathrm{TV}(P_\beta\|\hat{\pi})] \leq \sqrt{\frac{c_\beta^\star}{2} \mathbb{E}_{s \sim d_h^\beta}[D_{\mathrm{KL}}(P_\beta\|\hat{\pi})]} = \sqrt{\frac{c_\beta^\star}{2} L(\hat{\pi})}. \tag{46}$$

By MLE analysis (Liu et al., 2025), with probability $\geq 1 - \delta$:

$$L(\hat{\pi}) \leq \mathcal{O}\left(\sqrt{c \cdot \frac{\log |\Pi|/\delta}{N}}\right) + \delta_{\mathrm{approx}}. \tag{47}$$

Summing over $H$ stages:

$$\sum_{h=1}^H \mathbb{E}[\text{Term (I)}] \leq H\sqrt{\frac{c_\beta^\star}{2}} \left(\mathcal{O}\left(\left(\frac{\log |\Pi|/\delta}{N}\right)^{1/4}\right) + \sqrt{\delta_{\mathrm{approx}}}\right). \tag{48}$$

**Fourth, Bounding Term (II) of Equation (40).** By Assumption B.8, we have:

$$\mathrm{TV}(\hat{\pi}(\cdot|Q^\star)\|\hat{\pi}(\cdot|\hat{Q}^\tau)) \leq L_\pi|Q^\star - \hat{Q}^\tau|. \tag{49}$$

From Theorem 3.1, we have:

$$|Q^\star - \hat{Q}^\tau| \leq \epsilon_\tau + L_Q\sqrt{\epsilon_{\mathrm{NFs}}}. \tag{50}$$

Taking expectation under $d_h^{\pi_\beta^\star}$ and applying distribution mismatch for the NFs error term, we have:

$$\mathbb{E}_{s \sim d_h^{\pi_\beta^\star}}[|Q^\star - \hat{Q}^\tau|] \leq \epsilon_\tau + L_Q \cdot \mathbb{E}_{s \sim d_h^{\pi_\beta^\star}}[\sqrt{\epsilon_{\mathrm{NFs},s}}] \tag{51}$$

$$\leq \epsilon_\tau + L_Q\sqrt{c_\beta^\star \cdot \epsilon_{\mathrm{NFs}}}. \tag{52}$$

Summing over stages, we have:

$$\sum_{h=1}^H \mathbb{E}[\text{Term (II)}] \leq H \cdot L_\pi \left(\epsilon_\tau + L_Q\sqrt{c_\beta^\star \cdot \epsilon_{\mathrm{NFs}}}\right). \tag{53}$$

**Final Bound.** Combining, we have:

$$J(\pi_\beta^\star) - J(\hat{\pi}_{\mathcal{D}}^\star) \leq 2H\left[\sum_h \text{Term (I)} + \sum_h \text{Term (II)}\right] \tag{54}$$

$$\leq \mathcal{O}\left(\frac{c_\beta^\star H^2}{\tilde{c}}\sqrt{c}\left(\frac{\log |\Pi|/\delta}{N}\right)^{1/4}\right) + \sqrt{c_\beta^\star}H^2\sqrt{\delta_{\mathrm{approx}}} \tag{55}$$

$$+ c_\beta^\star H^2 L_\pi \left(\sqrt{c_\beta^\star \cdot \epsilon_{\mathrm{NFs}}} + \epsilon_\tau\right). \tag{56}$$

Union bound over events from Theorems 3.1 and MLE analysis gives probability $\geq 1 - 2\delta$. $\square$

## C.3. Comparison with $\text{R}^2\text{CSL}$

| Aspect | $\text{R}^2\text{CSL}$ | QHyer |
|---|---|---|
| Conditioning signal | RTG: $f(s, h) = \sum_{t=h}^{H} r_t$ | Q-value: $Q(s, a, g) = p_+^\beta(g|s, a)$ |
| Signal property | *Trajectory-dependent* | *state-action-dependent* |
| Consistency constraint | Required | **Not required** |
| Stitching mechanism | Explicit RTG relabeling | Implicit via expectile |
| Estimation method | Quantile regression | Expectile + NFs |
| Additional error term | None | $L_Q \sqrt{c_\beta^\star \epsilon_{\text{NFs}}}$ |
| Sample complexity | $|\mathcal{S}||\mathcal{A}|H$ | $|\mathcal{S}||\mathcal{A}||\mathcal{G}|H$ |
| Convergence rate | $\mathcal{O}(N^{-1/4})$ | $\mathcal{O}(N^{-1/4})$ |

# D. QHyer Algorithm Details

This section describes the architecture, training, and inference procedures of **QHyer**. The overall structure is depicted in Figure 8, and the complete algorithm is summarized in Algorithm 1.

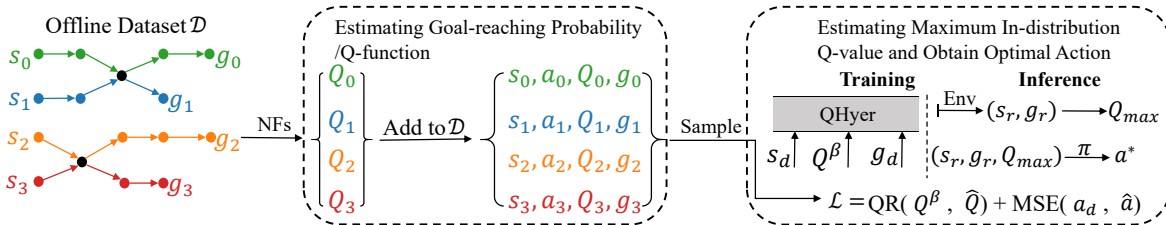

*Figure 8.* Overview of **QHyer**. **Left:** Offline dataset $\mathcal{D}$ with state-goal-action tuples. **Middle:** NFs-based critic estimates $Q_\theta^\beta(s_t, a_t, g) = \log p_\theta(g|s_t, a_t)$ via an SA-Encoder and RealNVP. **Right:** The Hybrid Attention-Mamba actor predicts $\hat{Q}(s_t, g)$ via expectile regression and outputs action $\hat{a}_t$ conditioned on the predicted maximum Q-value.

**Model Architecture.** The input sequence follows the format $\langle \tilde{Q}_t, sg_t, a_t \rangle$ where $sg_t = [s_t; g]$ denotes state-goal concatenation (Schaul et al., 2015), and $\tilde{Q}_t$ is the normalized Q-value computed from the NFs-based critic (Equation (4)):

$$\tilde{Q}_t = Q_\theta^\beta(s_t, a_t, g)/(\overline{|Q^\beta|} + \delta), \tag{57}$$

where $Q_\theta^\beta(s_t, a_t, g) = \log p_\theta(g|s_t, a_t)$ is the behavior Q-value estimated by NFs, $\overline{|Q^\beta|}$ denotes the mean absolute Q-value over the batch, and $\delta$ is a small constant for numerical stability. At timestep $t$, the model takes a context window of length $K$:

**Input:** $\langle \tilde{Q}_{t-K+1}, sg_{t-K+1}, a_{t-K+1}, \ldots, \tilde{Q}_t, sg_t, a_t \rangle$

**Output:** $\langle \hat{Q}_{t-K+1}, \hat{a}_{t-K+1}, \square, \ldots, \hat{Q}_t, \hat{a}_t, \square \rangle$

The NFs critic consists of an SA-Encoder that maps $(s_t, a_t)$ to a latent representation, followed by a coupling-based NFs (RealNVP) (Dinh et al., 2017) that computes $Q_\theta^\beta(s_t, a_t, g)$. The Hybrid Attention-Mamba backbone processes tokens through $N$ transformer blocks with learnable attention-Mamba gating as described in Section 3.2.

**Q-Conditioned Policy Learning.** Unlike prior Q-enhanced supervised learning methods that incorporate Q-values into loss functions, we use Q-values as **conditioning tokens** input to the policy network. This design enables the policy to explicitly leverage Q-value signals for action selection during both training and inference. The total loss is defined in Equation (14), combining the NFs-based critic loss (Equation (5)), behavior cloning loss (Equation (15)), and expectile regression loss (Equation (10)).

In practice, we apply a denoising trick to the NFs-based critic by adding Gaussian noise $\epsilon \sim \mathcal{N}(0, \sigma^2 I)$ to goals during training, which improves density estimation quality. The expectile loss $L_\tau^2(\cdot)$ is defined in Equation (9), where $\tau \in (0.5, 1)$

---

**Algorithm 1 QHyer** Training and Inference

---

1: **Input:** Offline dataset $\mathcal{D}$, context length $K$, expectile $\tau$, noise std $\sigma$, stability constant $\delta$
2: **Initialize:** SA-Encoder $\psi$, NFs-based critic $\theta$, Hybrid Attention-Mamba actor $\phi$
3:
4: // Joint Training (end-to-end)
5: **for** each training iteration **do**
6:     Sample batch of trajectories $\{(s_t, a_t, g)\}$ from $\mathcal{D}$
7:     // Step 1: Compute behavior Q-values via NF critic (Equation (4))
8:     $\text{repr}_t \leftarrow \text{SA-Encoder}_\psi(s_t, a_t)$
9:     $Q_\theta^\beta(s_t, a_t, g) \leftarrow \log p_\theta(g + \epsilon \mid \text{repr}_t)$                               // $\epsilon \sim \mathcal{N}(0, \sigma^2 I)$, denoising
10:     $\tilde{Q}_t \leftarrow Q_\theta^\beta(s_t, a_t, g)/(|Q^\beta| + \delta)$                                // normalize (Equation (57))
11:     // Step 2: Forward through Q-conditioned policy
12:     Construct input: $\mathbf{x} = \langle \tilde{Q}_{t-K+1}, sg_{t-K+1}, a_{t-K+1}, \ldots, \texttt{stopgrad}(\tilde{Q}_t), sg_t \rangle$
13:     $\hat{Q}(s_t, g), \hat{a}_t \leftarrow \text{QHyer}_\phi(\mathbf{x})$
14:     // Step 3: Compute losses (Equations (5), (10) and (15))
15:     $\mathcal{L}_{\text{NFs}} \leftarrow -\mathbb{E}[\log p_\theta(g|s_t, a_t)]$
16:     $\mathcal{L}_{\text{BC}} \leftarrow -\mathbb{E}[\log \pi_\phi(a_t|\tilde{Q}_t, [s_t; g])]$
17:     $\mathcal{L}_Q \leftarrow \mathbb{E}[L_\tau^2(\hat{Q}(s_t, g) - Q_\theta^\beta(s_t, a_t, g))]$
18:     Update $(\psi, \theta, \phi)$ by $\nabla(\lambda_{\text{critic}}\mathcal{L}_{\text{NFs}} + \lambda_{\text{BC}}\mathcal{L}_{\text{BC}} + \lambda_Q \mathcal{L}_Q)$
19: **end for**
20:
21: // Inference: Trajectory Stitching via Q-Conditioning
22: Initialize buffers: $\mathbf{sg}, \mathbf{a}, \mathbf{Q} \leftarrow \mathbf{0}$
23: $s_0, g \leftarrow \text{Env.reset}()$
24: **for** $t = 0, 1, 2, \ldots$ **until** done **do**
25:     $sg_t \leftarrow [s_t; g]$                                                // concatenate state and goal
26:     Retrieve context window: $(\mathbf{sg}, \mathbf{a}, \mathbf{Q})_{t-K+1:t}$
27:     // Stage 1: Predict maximum in-distribution Q-value
28:     $\hat{Q}(s_t, g) \leftarrow \text{QHyer}_\phi^Q(\mathbf{sg}_{t-K+1:t}, \mathbf{a}_{t-K+1:t-1}, \mathbf{Q}_{t-K+1:t-1})$
29:     Update buffer: $\mathbf{Q}_t \leftarrow \hat{Q}(s_t, g)$
30:     // Stage 2: Predict action conditioned on maximum Q-value
31:     $\hat{a}_t \leftarrow \text{QHyer}_\phi^a(\mathbf{sg}_{t-K+1:t}, \mathbf{a}_{t-K+1:t-1}, \mathbf{Q}_{t-K+1:t})$
32:     $s_{t+1} \leftarrow \text{Env.step}(\text{clip}(\hat{a}_t, -1, 1))$
33:     Update buffer: $\mathbf{a}_t \leftarrow \hat{a}_t$
34: **end for**

---

controls the asymmetry. When $\tau > 0.5$, overestimation is penalized more heavily, driving the learned $\hat{Q}(s_t, g)$ toward the maximum of $Q_\theta^\beta(s_t, a_t, g)$ over all actions in the dataset.

**Inference: Trajectory Stitching via Q-Conditioning.** In classical Q-learning, the optimal value function $Q^*$ derives the optimal action given the current state. In our framework, we leverage the maximum Q-value $\hat{Q}(s_t, g)$ to help the policy select near-optimal actions. Note that $\hat{Q}(s_t, g)$ depends only on state and goal because action is marginalized by the expectile regression. The inference pipeline follows:

$$\xmapsto{\text{Env}} (s_0, g) \xrightarrow{\hat{Q}} \hat{Q}(s_0, g) \xrightarrow{\pi_\phi} a_0 \xrightarrow{\text{Env}} (s_1, g) \xrightarrow{\hat{Q}} \hat{Q}(s_1, g) \xrightarrow{\pi_\phi} a_1 \rightarrow \cdots \tag{58}$$

At each timestep $t$, **QHyer** performs two-stage autoregressive generation as shown in Algorithm 1:

1. **Predict maximum Q-value:** Given the historical context window, the model first predicts $\hat{Q}(s_t, g)$ which represents the maximum achievable goal-reaching probability from the current state.

2. **Predict action:** Conditioned on the predicted $\hat{Q}(s_t, g)$, the model then outputs the action $\hat{a}_t$ that achieves this maximum Q-value.

When the initial state and goal correspond to different trajectories in the dataset, which is precisely the scenario requiring trajectory stitching, our model outputs effective actions by leveraging the Q-conditioned policy.

## E. Baseline Details

We compare our approach with a wide variety of baselines, including sequence modeling, TD-based RL methods and Offline GCRL methods. Particularly, we include the following methods:

- For sequence modeling methods, we include Decision Transformer (DT) (Chen et al., 2021), Elastic Decision Transformer (EDT) (Wu et al., 2023), Graph Decision Transformer (GDT) (Hu et al., 2023), Q-learning Decision Transformer (QDT) (Yamagata et al., 2023), Critic-Guided Decision Transformer (CGDT) (Wang et al., 2024), Reinforced Transformer (Reinformer) (Zhuang et al., 2024), Decision ConvFormer (DC) (Kim et al., 2024b), Decision Mamba (DMamba) (Ota, 2024), Q-value Regularized Transformer (QT) (Hu et al., 2024), Long-Short Decision Transformer (LSDT) (Wang et al., 2025), Decision Mixer (DMixer) (Zheng et al., 2025), Value-guided Decision Transformer (VDT) (Zheng et al., 2026). DT is a classic sequence modeling method that utilizes a Transformer architecture to model and reproduce sequences from demonstrations, integrating a goal-conditioned policy to convert Offline RL into a supervised learning task. Despite its competitive performance in Offline RL tasks, the DT falls short in achieving trajectory stitching (Brandfonbrener et al., 2022). GDT extends DT by explicitly structuring the input sequence as a causal graph and incorporating relation-enhanced attention to better model the dependencies between states, actions, and rewards. EDT is a variant of DT that lies in its ability to determine the optimal history length to promote trajectory stitching. But it does not incorporate the RL objective that maximizes returns to enhance the model (Zhuang et al., 2024) and its stitching capabilities are limited (Kim et al., 2024a). QDT integrates Dynamic Programming with the DT framework to enhance the optimal path generation ability of DT. CGDT enhances DT by incorporating a value-based critic to align the expected returns of actions with target returns, effectively addressing the inconsistency issues of Return-Conditioned Supervised Learning in stochastic environments and suboptimal datasets. DC replaces attention blocks with convolution filters to more efficiently capture local associations. Reinformer is similar to our work; however, it exhibits limited stitching capabilities due to the absence of $Q$-value, resulting in a significant performance gap compared to TD-based RL methods. DMamba replaces the attention mechanism in DT with the Mamba selective state space model to achieve linear computational complexity while maintaining sequence modeling capabilities. QT introduces Q-value regularization to optimize action selection on top of DT and excels in handling long time horizons and sparse reward tasks. LSDT enhances the model structure of DT with a dual-branch architecture (long-term and local features) adept at extracting information within different ranges. DMixer integrates both long-term and local features, and additionally introduces a plug-and-play dynamic token selection mechanism to ensure that the model can adaptively allocate attention to different features based on the specific requirements of each task. VDT leverages value functions to perform advantage-weighting and behavior regularization on the DT, guiding the policy toward upper-bound optimal decisions during the offline training phase.

- For TD-based RL methods, we include Conservative Q-Learning (CQL) (Kumar et al., 2020) and Implicit Q-Learning (IQL) (Kostrikov et al., 2022). CQL and IQL are classical offline RL methods that utilize dynamic programming. This trick endows them with stitching properties (Cheikhi & Russo, 2023; Ghugare et al., 2024).

- For Offline GCRL methods, we include goal-conditioned behavioral cloning (GCBC) (Ghosh et al., 2021), goal-conditioned implicit V-learning (GCIVL) and Q-learning (GCIQL) (Kostrikov et al., 2022), Quasimetric RL (QRL) (Wang et al., 2023), Contrastive RL (CRL) (Eysenbach et al., 2022), and Hierarchical implicit Q-learning (HIQL) (Park et al., 2023). For these baselines, we follow the implementation setup established by OGBench (Park et al., 2025) throughout our experiments. Additionally, we select Subgoal Advantage-Weighted Policy Bootstrapping (SAW) (Zhou & Kao, 2026), Option-aware Temporally Abstracted (OTA) (Ahn et al., 2026) and Eikonal-Constrained Quasimetric RL (Eik-HiQRL) (Giammarino & Qureshi, 2026) as our state-of-the-art GCRL baselines. SAW trains a flat policy by directly sampling subgoals from offline datasets through advantage-weighted policy bootstrapping, thereby eliminating the need for complex subgoal generation models, and achieves superior performance on long-horizon, high-dimensional control tasks. OTA employs temporal abstraction to reduce the effective planning horizon, which substantially improves the scalability of high-level policies to long-horizon tasks. Eik-HiQRL overcomes QRL's dependence on trajectory continuity for local constraints and its struggle to maintain a valid quasimetric structure in high-dimensional, long-horizon tasks by introducing a trajectory-free Eikonal PDE constraint at the high level and a hierarchical policy decomposition.

# F. Experiment Details

In this section we provide offline datasets details as well as implementation details used for all the algorithms in our experiments – Offline GCRL Datasets, NFs, and **QHyer**.

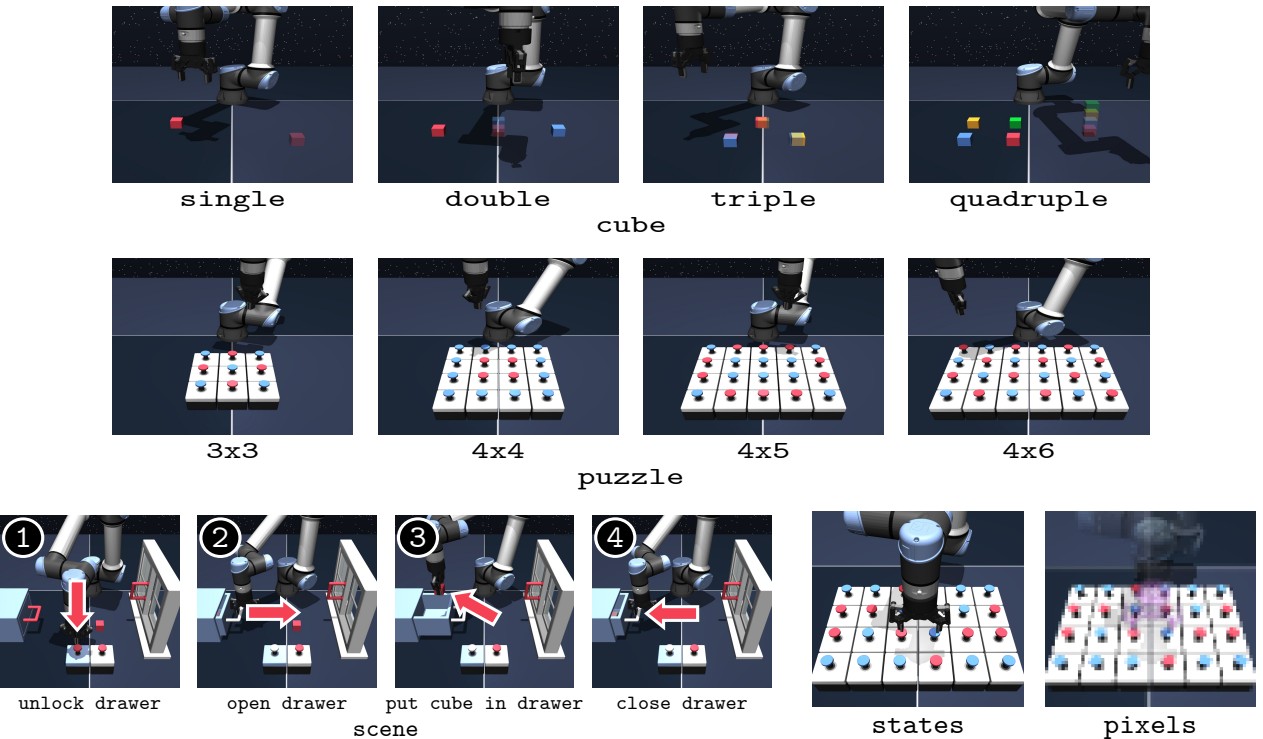

*Figure 9.* **GCRL example non-Markovian datasets from Ogbench.** Each Trajectory is limited to travel at most 4 blocks for dataset type stitch, while at inference, the distance between the start and goal can be up to 30 in the Giant maze.

## F.1. Offline GCRL non-Markovian Datasets

We adopt the manipulation suite from **OGBench** (Park et al., 2025), which consists of three robotic manipulation environments based on a 6-DoF UR5e robot arm. These environments are designed to evaluate the agent's capabilities in object manipulation, sequential generalization, and combinatorial generalization.

- Cube: This task involves pick-and-place manipulation of cube blocks, where the goal is to arrange cubes into designated configurations. Four variants are provided with different numbers of cubes: single, double, triple, and quadruple (1–4 cubes). At test time, the agent must perform moving, stacking, swapping, or permuting operations on the cube blocks.

- Scene: This task is designed to challenge sequential, long-horizon reasoning capabilities. It involves manipulating diverse everyday objects including a cube block, a window, a drawer, and two button locks. The longest evaluation task requires completing up to eight atomic behaviors in sequence.

- Puzzle: This task evaluates combinatorial generalization by requiring the agent to solve the "Lights Out" puzzle with a robot arm. Four difficulty levels are provided: 3x3, 4x4, 4x5, and 4x6, with state spaces containing up to $2^{24} = 16,777,216$ distinct configurations.

Visualization examples of these tasks are shown in Figure 9. For each manipulation environment, OGBench provides two types of datasets with different collection policies:

- **Play datasets** (play): Collected by non-Markovian expert policies with temporally correlated noise, following the

"play data" paradigm (Lynch et al., 2020). This results in smoother, more realistic trajectories that pose additional challenges for standard RL algorithms.

- **Noisy datasets** (`noisy`): Collected by Markovian expert policies with uncorrelated Gaussian noise. These datasets serve as controlled baselines for ablation studies, allowing researchers to isolate the effects of non-Markovian data collection.

In the experiments comparing with related sequence modeling approaches, we adopt the maze navigation tasks from **D4RL** (Fu et al., 2020), which provide challenging benchmarks for evaluating offline RL algorithms on undirected, multitask data with sparse rewards.

- `Maze2D`: This domain is a navigation task requiring a 2D point-mass agent to reach a fixed goal location. Three maze layouts are provided with increasing complexity: `umaze`, `medium`, and `large`. The tasks are designed to test the ability of offline RL algorithms to stitch together previously collected sub-trajectories to find the shortest path to the evaluation goal.

- `AntMaze-v2`: This domain replaces the simple 2D ball from Maze2D with a more complex 8-DoF quadrupedal "Ant" robot, introducing morphological complexity that mimics real-world robotic navigation tasks. The same three maze layouts (`umaze`, `medium`, `large`) are used, with a sparse 0-1 reward that is activated only upon reaching the goal. Three dataset variants are provided: standard goal-reaching from fixed start locations, "diverse" datasets with random start and goal locations, and "play" datasets with hand-picked navigation waypoints.

Visualization examples are shown in Figure 10. A critical characteristic of both `Maze2D` and `AntMaze-v2` datasets is that they are collected by non-Markovian policies. The data generation process employs a hierarchical controller: a high-level planner generates sequences of waypoints, which are then followed by a low-level PD controller (for `Maze2D`) or a trained goal-reaching policy (for `AntMaze-v2`). Because these controllers maintain internal states to track visited waypoints and update their targets upon reaching intermediate goals, the resulting behavior policies are inherently non-Markovian. This property introduces additional challenges for offline RL algorithms, as the data cannot be accurately modeled by assuming a Markovian behavior policy, potentially causing bias in methods that rely on such assumptions (Fu et al., 2020).

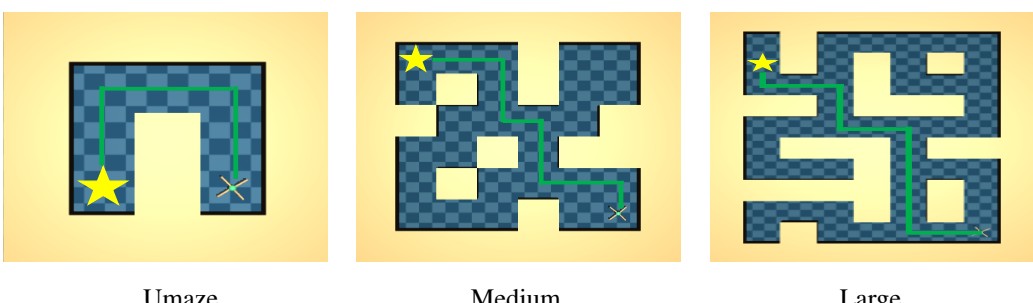

| Umaze | Medium | Large |

*Figure 10.* **GCRL example non-Markovian datasets from D4RL** (Fu et al., 2020): The `AntMaze-v2` datasets involve controlling an 8-DoF quadruped to navigate towards a specified goal state. This benchmark requires value propagation to effectively stitch together sub-optimal trajectories from the collected data.

### F.2. Implementation Details

We ran all our experiments on NVIDIA RTX 3090 GPUs with 24GB of memory within an internal cluster. We use the default configurations in Park et al. (2025), with some values modified. In pixel-based environments, following Park et al. (2025), we employ a IMPALA-style encoder to transform images into state tokens. The architecture and training process of the NFs are identical to those described in Ghugare & Eysenbach (2026).

Our **QHyer** implementation draws inspiration from LSDT (Wang et al., 2025) and Decision Mamba (Ota, 2024). The state tokens, goal tokens, $Q$-value tokens and action tokens are first processed by different linear layers. Then these tokens are fed into the decoder layer to obtain the embedding. Here the decoder layer is a lightweight implementation from Reinformer (Zhuang et al., 2024). The context length for the decoder layer is denoted as $K$. We employ the AdamW (Loshchilov & Hutter, 2019) optimizer to optimize the total loss, in alignment with the methods outlined in their original papers. The expectile parameter of $L_{QHyer}$ loss is denoted as $\tau$.

*Table 5.* NF Q-value Estimation Time Analysis in **QHyer**. All timing values are in milliseconds (ms) per batch, averaged over 100 steps (batch size = 256). Experiments conducted on a single NVIDIA RTX 3090 GPU with dual Intel Xeon E5-2620 v4 CPUs @ 2.10GHz.

| Environment | NF Train (ms) | Actor (ms) | Infer-Q (ms) | Infer-A (ms) | NFs Ratio |
|---|---|---|---|---|---|
| `cube-single-play-v0` | 2.42 | 3.07 | 0.007 | 0.015 | 28.3% |
| `cube-double-play-v0` | 2.43 | 2.99 | 0.008 | 0.019 | 26.7% |
| `cube-triple-play-v0` | 2.47 | 3.14 | 0.008 | 0.015 | 28.8% |
| `cube-quadruple-play-v0` | 2.54 | 3.12 | 0.011 | 0.019 | 27.0% |
| `cube-single-noisy-v0` | 2.44 | 3.06 | 0.009 | 0.015 | 29.3% |
| `cube-double-noisy-v0` | 2.49 | 3.07 | 0.011 | 0.019 | 26.9% |
| `cube-triple-noisy-v0` | 2.42 | 3.01 | 0.008 | 0.014 | 29.2% |
| `cube-quadruple-noisy-v0` | 2.55 | 3.14 | 0.012 | 0.018 | 27.1% |
| `scene-play-v0` | 2.43 | 3.04 | 0.009 | 0.015 | 28.8% |
| `scene-noisy-v0` | 2.43 | 3.02 | 0.008 | 0.020 | 26.9% |
| `puzzle-3x3-play-v0` | 2.39 | 3.01 | 0.008 | 0.015 | 28.7% |
| `puzzle-4x4-play-v0` | 2.52 | 3.08 | 0.010 | 0.019 | 27.1% |
| `puzzle-4x5-play-v0` | 2.48 | 3.12 | 0.008 | 0.015 | 28.7% |
| `puzzle-4x6-play-v0` | 2.54 | 3.10 | 0.009 | 0.020 | 26.8% |
| `puzzle-3x3-noisy-v0` | 2.45 | 3.10 | 0.009 | 0.015 | 29.0% |
| `puzzle-4x4-noisy-v0` | 2.49 | 3.09 | 0.009 | 0.019 | 26.9% |
| `puzzle-4x5-noisy-v0` | 2.51 | 2.95 | 0.009 | 0.015 | 30.1% |
| `puzzle-4x6-noisy-v0` | 2.44 | 2.98 | 0.008 | 0.021 | 27.2% |
| **Average** | 2.47 | 3.06 | 0.009 | 0.017 | 28.0% |

## F.3. Hyperparameter Settings

Table 6 summarizes the hyperparameters shared across all experiments. The Hybrid Attention-Mamba architecture uses learnable mixing weights between attention and Mamba branches, with a total hidden dimension of $d_{model}$. The Normalizing Flow architecture follows Ghugare & Eysenbach (2026). The expectile regression parameter $\tau$ is set according to our theoretical guidance (Theorem 3.1).

*Table 6.* Common hyperparameters shared across all tasks.

| Hyperparameter | OGBench (State) | OGBench (Pixel) | D4RL |
|---|---|---|---|
| Training steps | 1M | 500K | 100K |
| Batch size | 1024 | 512 | 256 |
| Optimizer | AdamW | AdamW | AdamW |
| Weight decay | 0.0 | 0.0 | 1e-4 |
| Gradient clipping | 1.0 | 1.0 | 0.25 |
| NF noise std | 0.05 | 0.05 | – |
| Encoder hidden dim | 1024 | 1024 | – |
| NF representation size | 64–128 | 64–128 | – |
| BC weight $\lambda_{BC}$ | 1.0 | 1.0 | 1.0 |
| Q weight $\lambda_Q$ | 1.0 | 1.0 | 1.0 |
| Expectile $\tau$ | 0.95–0.99 | 0.95–0.99 | 0.90–0.99 |
| State-goal concatenation | True | False | True |
| Image size | – | 64×64 | – |
| Image encoder | – | IMPALA-small | – |
| Warmup steps | – | – | 10000 |
| LR schedule | – | – | Cosine |

**Architecture notes.** For both OGBench and D4RL experiments, the Hybrid Attention-Mamba backbone uses *learnable mixing weights* that are automatically optimized during training, eliminating the need for manual tuning of attention-to-Mamba ratios. The total hidden dimension $d_{model}$ (denoted as `h_dim` in OGBench and `embed_dim` in D4RL) represents the combined capacity of both branches, with the proportion learned end-to-end via gradient descent.

Table 7 presents the task-specific hyperparameters for OGBench state-based manipulation environments. Following our theoretical analysis (Theorem 3.1), we set $\tau = 0.99$ for play datasets (medium Q-value coverage) and $\tau = 0.95$ for noisy datasets (higher coverage due to exploration noise).

Table 8 presents the hyperparameters for pixel-based (visual) manipulation tasks. Compared to state-based tasks, pixel-based tasks use smaller batch size (512 vs 1024) due to memory constraints and shorter training (500K steps). Goals are represented

*Table 7.* Task-specific hyperparameters for OGBench state-based manipulation tasks (1M training steps). $K$: context length, $d_{\mathrm{model}}$: hidden dimension, $L$: number of Transformer blocks, $H$: number of attention heads.

| Environment | $K$ | $d_{\mathbf{model}}$ | $L$ | $H$ | **LR** | **Dropout** | $\tau$ | **NF Blocks** | **NF Channels** |
|---|---|---|---|---|---|---|---|---|---|
| cube-single-play-v0 | 20 | 256 | 4 | 4 | 3e-4 | 0.1 | 0.99 | 6 | 256 |
| cube-single-noisy-v0 | 20 | 256 | 4 | 4 | 3e-4 | 0.1 | 0.95 | 6 | 256 |
| cube-double-play-v0 | 25 | 384 | 5 | 6 | 3e-4 | 0.1 | 0.99 | 8 | 256 |
| cube-double-noisy-v0 | 25 | 384 | 5 | 6 | 3e-4 | 0.1 | 0.95 | 8 | 256 |
| cube-triple-play-v0 | 30 | 512 | 6 | 8 | 2e-4 | 0.15 | 0.99 | 10 | 384 |
| cube-triple-noisy-v0 | 30 | 512 | 6 | 8 | 2e-4 | 0.15 | 0.95 | 10 | 384 |
| cube-quadruple-play-v0 | 35 | 640 | 6 | 8 | 1e-4 | 0.2 | 0.99 | 12 | 512 |
| cube-quadruple-noisy-v0 | 35 | 640 | 6 | 8 | 1e-4 | 0.2 | 0.95 | 12 | 512 |
| scene-play-v0 | 30 | 384 | 5 | 6 | 3e-4 | 0.1 | 0.99 | 8 | 384 |
| scene-noisy-v0 | 30 | 384 | 5 | 6 | 3e-4 | 0.1 | 0.95 | 8 | 384 |
| puzzle-3x3-play-v0 | 25 | 512 | 6 | 8 | 3e-4 | 0.1 | 0.99 | 8 | 384 |
| puzzle-3x3-noisy-v0 | 25 | 512 | 6 | 8 | 3e-4 | 0.1 | 0.95 | 8 | 384 |
| puzzle-4x4-play-v0 | 30 | 640 | 6 | 8 | 2e-4 | 0.15 | 0.99 | 10 | 384 |
| puzzle-4x4-noisy-v0 | 30 | 640 | 6 | 8 | 2e-4 | 0.15 | 0.95 | 10 | 384 |
| puzzle-4x5-play-v0 | 35 | 768 | 6 | 8 | 1e-4 | 0.2 | 0.99 | 10 | 512 |
| puzzle-4x5-noisy-v0 | 35 | 768 | 6 | 8 | 1e-4 | 0.2 | 0.95 | 10 | 512 |
| puzzle-4x6-play-v0 | 40 | 768 | 6 | 8 | 1e-4 | 0.2 | 0.99 | 10 | 512 |
| puzzle-4x6-noisy-v0 | 40 | 768 | 6 | 8 | 1e-4 | 0.2 | 0.95 | 10 | 512 |

as images rather than concatenated state vectors.

*Table 8.* Task-specific hyperparameters for OGBench pixel-based manipulation tasks (500K training steps). $K$: context length, $d_{\mathrm{model}}$: hidden dimension, $L$: number of Transformer blocks, $H$: number of attention heads.

| Environment | $K$ | $d_{\mathbf{model}}$ | $L$ | $H$ | **LR** | **Dropout** | $\tau$ | **NF Blocks** | **NF Channels** |
|---|---|---|---|---|---|---|---|---|---|
| visual-cube-single-play-v0 | 15 | 256 | 4 | 4 | 3e-4 | 0.1 | 0.99 | 6 | 256 |
| visual-cube-double-play-v0 | 20 | 384 | 5 | 6 | 3e-4 | 0.1 | 0.99 | 8 | 256 |
| visual-cube-triple-play-v0 | 25 | 512 | 6 | 8 | 2e-4 | 0.15 | 0.99 | 10 | 384 |
| visual-scene-play-v0 | 25 | 384 | 5 | 6 | 3e-4 | 0.1 | 0.99 | 8 | 384 |
| visual-scene-noisy-v0 | 25 | 384 | 5 | 6 | 3e-4 | 0.1 | 0.95 | 8 | 384 |

For pixel-based tasks, the NF uses a DrQ-v2 style CNN (Yarats et al., 2022) to encode images into 256-dim features, which are concatenated with actions and passed through a 4-layer MLP to produce the state-action representation. The NF models goal-reaching probability in the low-dimensional coordinate space (e.g., object positions), extracted from simulator state. The LSDM actor uses an IMPALA-style encoder (Espeholt et al., 2018) to encode both observation and goal images into 256-dim vectors.

Table 9 presents the hyperparameters for D4RL maze tasks. We use a unified Transformer architecture with $d_{\mathrm{model}} = 128$ and 3 blocks. Unlike OGBench, D4RL experiments use a cosine learning rate schedule with 10K warmup steps.

*Table 9.* Task-specific hyperparameters for D4RL maze tasks (100K training steps, 50 iterations $\times$ 2000 updates).

| Environment | $K$ | **LR** | $\tau$ | $d_{\mathbf{model}}$ | $L$ |
|---|---|---|---|---|---|
| antmaze-umaze-v2 | 2 | 2e-4 | 0.90 | 128 | 3 |
| antmaze-umaze-diverse-v2 | 2 | 2e-4 | 0.90 | 128 | 3 |
| antmaze-medium-play-v2 | 3 | 2e-4 | 0.99 | 128 | 3 |
| antmaze-medium-diverse-v2 | 3 | 2e-4 | 0.99 | 128 | 3 |
| antmaze-large-play-v2 | 3 | 4e-4 | 0.90 | 128 | 3 |
| antmaze-large-diverse-v2 | 3 | 4e-4 | 0.90 | 128 | 3 |
| maze2d-umaze-v1 | 10 | 2e-4 | 0.90 | 128 | 3 |
| maze2d-medium-v1 | 10 | 2e-4 | 0.90 | 128 | 3 |

**D4RL-specific settings.** For D4RL maze tasks, we concatenate the 2D goal position to the state (-goalconcate), increasing the state dimension by 2. The training uses a combined learning rate schedule: linear warmup for 10K steps followed by cosine decay. We use smaller batch size (256) and fewer training steps (100K) compared to OGBench, as D4RL maze tasks are less complex. The expectile parameter $\tau$ is set to 0.90 for umaze and large tasks, and 0.99 for medium tasks based on empirical tuning.

For computational efficiency, we extract only task-relevant goal coordinates when training the NFs-based Q-value estimator in Equation (5). Given a full goal state $g_{\mathrm{full}}$, we use $g = g_{\mathrm{full}}[i_{\mathrm{start}} : i_{\mathrm{end}}]$ where the index range is environment-specific.

Table 10 summarizes the configurations:

*Table 10.* Goal dimension configurations for NF Q-value estimation.

| Task Category | Goal Dim | Description |
|---|---|---|
| `cube-single-*`/`visual-cube-single-*` | 3 | Object (x, y, z) |
| `cube-double-*`/`visual-cube-double-*` | 6 | Two objects |
| `cube-triple-*`/`visual-cube-triple-*` | 9 | Three objects |
| `cube-quadruple-*` | 12 | Four objects |
| `scene-*`/`visual-scene-*` | 13 | Scene objects |
| `puzzle-3x3-*` | 9 | 3×3 tiles |
| `puzzle-4x4-*` | 16 | 4×4 tiles |
| `puzzle-4x5-*` | 20 | 4×5 tiles |
| `puzzle-4x6-*` | 24 | 4×6 tiles |
| `antmaze-*`, `maze2d-*` | 2 | Agent (x, y) |

For goal sampling in OGBench, we use $p_{\text{trajgoal}} = 1.0$, $p_{\text{randomgoal}} = 0.0$ for play datasets and $p_{\text{trajgoal}} = 0.8$, $p_{\text{randomgoal}} = 0.2$ for noisy datasets.

## G. Additional Results

This section presents supplementary experiments and analyses for **QHyer**, including: (1) detailed discussion of the state-goal tokenization strategy and its role in enabling trajectory stitching, (2) ablation studies on regression functions, (3) qualitative visualization of trajectory stitching capabilities, (4) validation of NFs for goal-reaching probability estimation, and (5) empirical verification of expectile regression for capturing maximum Q-values. Due to space constraints, these additional results are not included in the main body of this paper. The details are provided below.

### G.1. Detail Discussion of State-Goal Tokenization Strategy

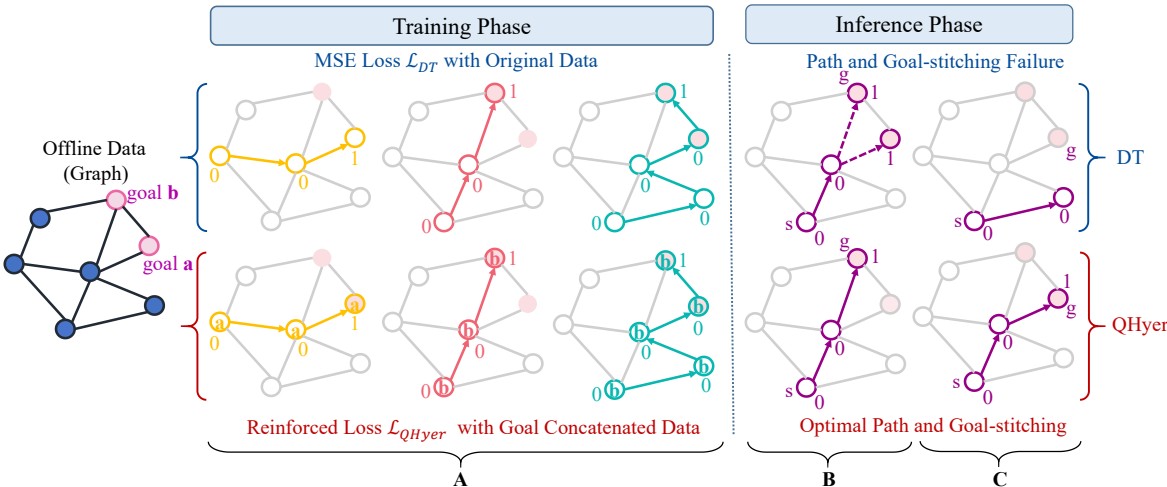

*Figure 11.* **State-goal tokenization strategy.** **(A)** Offline data represented as a graph, where nodes denote states and edges represent transitions. Different trajectories (colored in orange/yellow) may share common states but target different goals. **(B)** State-goal concatenation: each state $s$ is concatenated with goal $g$ to form a unified token $[s; g]$, enabling the model to directly attend to goal-relevant state features. **(C)** Goal stitching illustration (see Section 3.3): by conditioning on concatenated state-goal tokens, QHyer can identify and combine successful trajectory segments from different source trajectories that share the same goal, enabling optimal path discovery that neither original trajectory achieves alone.

This section details our state-goal tokenization strategy illustrated in Figure 11 and its role in enabling trajectory stitching. The key insight is that concatenating state and goal into a unified token $[s; g]$ allows the Transformer's self-attention mechanism to directly model cross-dependencies between current state features and goal specifications within each token position.

**Panel A** shows the offline dataset structure as a graph, where multiple trajectories (indicated by different colors) traverse

overlapping state regions while pursuing different goals. This shared structure creates opportunities for trajectory stitching, which combines successful segments from different trajectories.

**Panel B** contrasts DT's standard tokenization with QHyer's approach. In vanilla DT, states and goals may be processed separately or with weak coupling. QHyer instead concatenates $[s; g]$ at each timestep, ensuring that goal information is directly available when computing attention over state features. This design maintains the sequence length at $3T$ (Q-value, state-goal, action tokens) rather than increasing to $4T$ with separate goal tokens, avoiding quadratic attention overhead.

**Panel C** demonstrates how this enables goal stitching. Consider two trajectories targeting goals $a$ and $b$ respectively. Neither trajectory alone reaches the optimal path to goal $a$. However, by conditioning on state-goal concatenated tokens with NFs-based Q-value signals, QHyer identifies high-value segments from both trajectories and stitches them together, discovering an optimal path (shown in green) that was not present in any single demonstration.

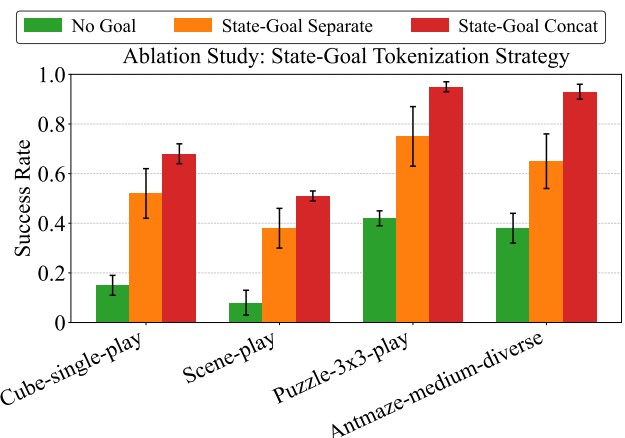

*Figure 12.* Ablation study on state-goal tokenization strategies.

We empirically validate the effectiveness of state-goal concatenation through ablation studies comparing three tokenization strategies: **No Goal** (state-only input), **State-Goal Separate** (goal as additional token), and **State-Goal Concat** (our approach). Figure 12 shows a consistent ordering across all environments: No Goal < Separate < Concat.

The performance gap between No Goal and goal-conditioned variants (37%–53% absolute improvement) confirms that goal information is essential for learning meaningful goal-reaching behaviors. Without explicit goal conditioning, the model degenerates to unconditional behavior cloning, unable to distinguish between trajectories targeting different goals.

Among goal-conditioned strategies, concatenation outperforms separation by 16%–28%. This improvement stems from two factors: (1) **Direct cross-dependency modeling**: Concatenation enables self-attention to directly learn which state features are relevant for specific goals within each token, whereas separation requires the model to establish state-goal relationships across tokens through multiple attention layers. (2) **Stronger conditioning signal**: Separate tokenization dilutes the goal signal as it propagates through attention layers, weakening goal-awareness at decision time. Concatenation preserves the full goal information at every position where action prediction occurs.

These results validate our design choice and explain why QHyer achieves effective trajectory stitching: the concatenated state-goal representation provides the necessary goal-aware context for identifying and combining high-value segments from different trajectories.

### G.2. Effect of Regression Functions on Learning Stability

We compare MSE, Quantile Loss ($L_1$-based) (Koenker & Hallock, 2001), and Expectile Regression ($L_2$-based) (Newey & Powell, 1987; Kostrikov et al., 2022). Figure 13 shows consistent ordering: MSE < Quantile < Expectile, with Expectile achieving the best results and smallest variance.

**Why MSE Fails.** MSE learns the mean Q-value across all trajectories passing through each state. In Offline GCRL where both successful and failed trajectories share common states, this averaging produces predictions that lie between the maximum and minimum Q-values. Such middle-ground estimates

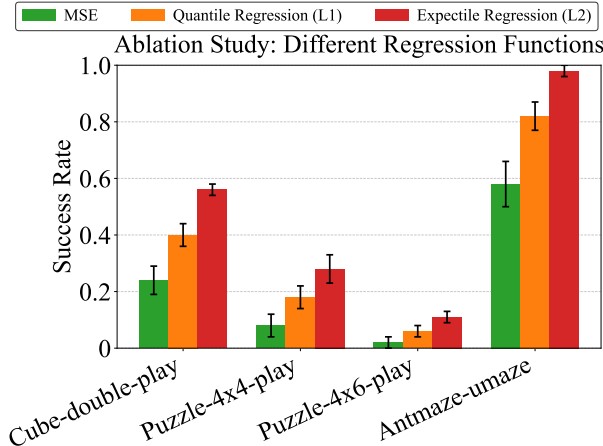

*Figure 13.* Ablation study on regression functions for maximum Q-value learning.

provide no discriminative signal for trajectory stitching because the model cannot distinguish promising paths from dead ends.

**Why Quantile Loss Struggles.** Quantile regression (Koenker & Hallock, 2001) correctly targets high-value regions via asymmetric weighting. However, the $L_1$ loss creates a non-smooth point at zero error where gradients change direction abruptly (Liu et al., 2025; Jullien et al., 2025). For deep networks with many near-zero predictions, this causes oscillatory training dynamics and high variance across seeds. Recent theoretical work (Liu et al., 2025) shows that while quantile regression can recover in-distribution optimal values in deterministic environments, its $L_1$ loss makes optimization less stable than $L_2$-based alternatives.

**Why Expectile Regression Succeeds.** Expectile regression (Newey & Powell, 1987) replaces the $L_1$ non-smooth point with an $L_2$ smooth curve, achieving both optimistic targeting and gradient consistency. This smooth gradient landscape is particularly important for non-Markovian learning: inconsistent gradients from quantile loss disrupt the temporal representations learned by attention and Mamba branches, while expectile's stable gradients allow these components to capture history-dependent patterns effectively. This explains why the Quantile-Expectile gap is largest on `Cube-double-play`, the environment with the strongest non-Markovian properties. As shown in Theorem 3.1, expectile regression with $\tau \to 1$ converges to the in-distribution optimal Q-value, providing theoretical justification for our empirical findings.

### G.3. Trajectory Stitching Visualization

To further illustrate the trajectory stitching capabilities of different methods, we provide a qualitative comparison on the D4RL `Antmaze-Medium` task. As shown in Figure 15, we visualize the trajectories generated by DT, LSDT, IQL, and **QHyer** (with Expectile Regression).

The maze environment consists of multiple regions, each represented by a distinct color corresponding to different data collection policies in the offline dataset (Figure 14):

- **Cyan**: Bottom-left start region

- **Purple**: Middle corridor

- **Yellow**: Top-right goal region

- **Green**: Bottom-right area

- **Red**: Top-left area

- **Black**: Out-of-distribution (OOD) states (i.e., passing through walls)

*Figure 14.* D4RL `Antmaze-medium` environment with trajectories from different behavioral policies.

The key challenge is to stitch trajectory segments from different regions to discover optimal paths from start to goal.

| (a) DT | (b) LSDT | (c) IQL | (d) QHyer (Ours) |
|---|---|---|---|
| 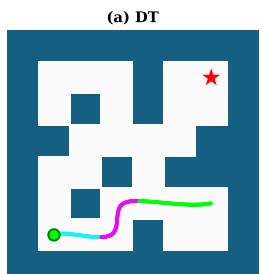 | 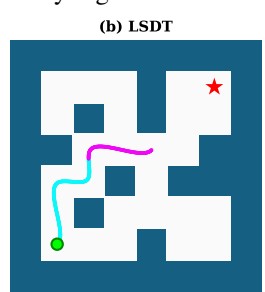 | 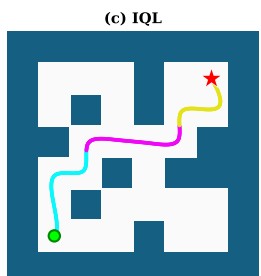 | 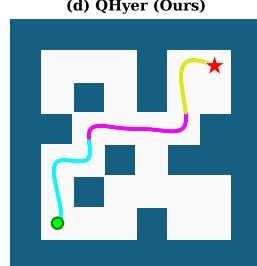 |

*Figure 15.* Qualitative comparison of trajectory stitching capabilities on D4RL `Antmaze-Medium` task. Different colors represent trajectory segments from different data collection policies in the offline dataset. **Black segments indicate OOD states** where the agent passes through walls. (a) DT fails to reach the goal due to ineffective RTG conditioning. (b) LSDT moves correctly but stops early. (c) IQL successfully reaches the goal via value-based stitching, but requires bootstrapping and policy projection. (d) **QHyer** successfully reaches the goal with the advantages of **no bootstrapping**, **no policy projection**, and **sequence modeling for non-Markovian data**.

Successful trajectory stitching requires the agent to combine trajectory segments from different regions to reach the goal. Our key observations are:

- **DT** (Chen et al., 2021) fails to reach the goal and instead wanders toward the bottom-right area, demonstrating its inability to stitch trajectories across different data collection policies. This failure stems from DT's reliance on

return-to-go conditioning, which provides no discriminative signal in sparse reward settings where all failed trajectories receive identical RTG values.

- **LSDT** (Wang et al., 2025) moves in the correct direction but stops in the middle corridor, showing limited stitching capability. Although LSDT improves upon DT by combining attention with Dynamic Convolution for better local pattern extraction, it still relies on RTG conditioning and cannot identify high-value stitching points without explicit value guidance.

- **IQL** (Kostrikov et al., 2022) successfully reaches the goal through a valid path without OOD states. IQL's expectile regression-based value learning enables trajectory stitching by identifying high-value actions. However, IQL requires bootstrapping to learn the maximum Q-value, which means it must first learn $Q^\beta$ before learning $\hat{Q}$. This can lead to error accumulation in complex environments.

- **QHyer** also successfully reaches the goal through a valid path without any OOD states. The trajectory smoothly transitions through cyan $\rightarrow$ purple $\rightarrow$ yellow regions, demonstrating proper trajectory stitching. Compared to IQL, QHyer avoids bootstrapping by using NFs for direct Q-value estimation, and avoids policy projection by using Q-conditioned supervised learning.

### G.4. Evaluating the Capability of NFs to Accurately Estimate Goal-reaching Probability

In this section, we validate the accuracy of the NFs's (Ghugare & Eysenbach, 2026) estimation of the discounted future state distribution by implementing the computation method outlined in Eysenbach et al. (2021) within a tabular setting. It is important to note that here we are solely validating the accuracy of the NFs in estimating the discounted future state distribution, which is unrelated to the actual implementation of the NFs in our **QHyer** framework.

Specifically, we compute the true discounted future state distribution in a modified GridWorld environment example and evaluate the estimation error by comparing it against the true distribution. We also compare the predictions of CVAE(Sohn et al., 2015), C-learning (Eysenbach et al., 2021) and CRL(Eysenbach et al., 2022) with the true future state density. First, we introduce the modified GridWorld environment used in this experiment. This environment is characterized by stochastic dynamics and a continuous state space, such that the true $Q$-function for the indicator reward is zero. Specifically, the environment has a size of $5 \times 5$ (Figure 16), where the agent observes a noisy version of its current state. More precisely, when the agent is located at position $(i, j)$, it observes the state $(i + \epsilon_i, j + \epsilon_j)$, where $\epsilon_i, \epsilon_j \sim \text{Unif}[-0.5, 0.5]$. Note that the observation uniquely identifies the agent's position, so there is no partial observability. Similar to Eysenbach et al. (2021), we analytically compute the exact future state density function by first determining the future state density of the underlying GridWorld, noting that the density is uniform within each cell. We generated a tabular policy by sampling from a Dirichlet (1) distribution, and sampled 100 trajectories of length 100 from this policy for NFs training.

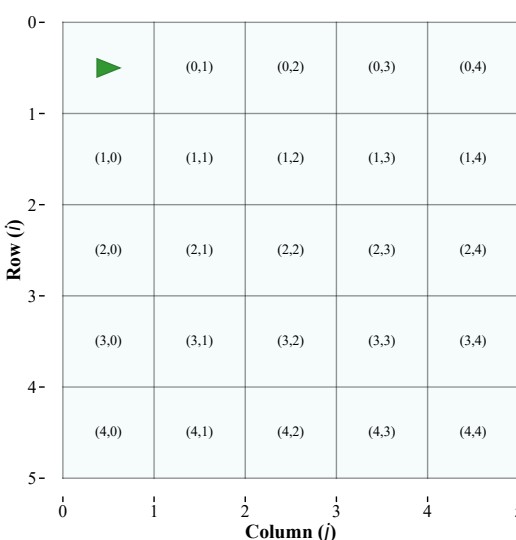

*Figure 16.* $5 \times 5$ Gridworld environment.

**Analytic Future State Distribution**    Then, as described in Eysenbach et al. (2021), we can compute the true discounted future state distribution by first constructing the following two metrics:

$$T \in \mathbb{R}^{25 \times 25} : \quad T[s, s'] = \sum_a \mathbb{1}(f(s, a) = s')\pi(a \mid s)$$

$$T_0 \in \mathbb{R}^{25 \times 4 \times 25} : \quad T[s, a, s'] = \mathbb{1}(f(s, a) = s'),$$

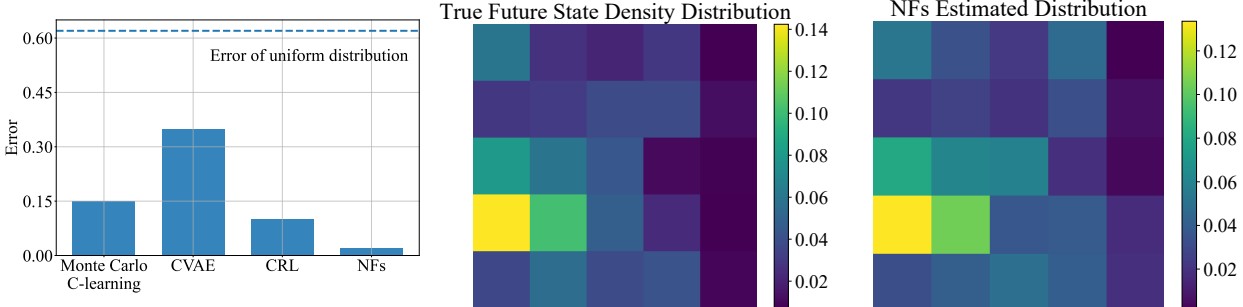

*Figure 17.* **Experiments on the effectiveness of density estimation using NFs. Left:** We evaluate CVAE, C-learning, CRL and NFs for predicting the future state distribution in the on-policy setting. As anticipated, NFs demonstrated the lowest estimation error among all methods evaluated. Conversely, CVAE exhibited the poorest estimation accuracy. In our empirical implementation, we observed that CVAE incurs significantly higher computational complexity due to its requirements for pre-training and importance sampling-based inference procedures (Wu et al., 2022). **Middle:** and **Right:** The visual comparison. For a given state, action, and future goal in the GridWorld trajectory data, we visualize the comparison between the actual future state density (goal-reaching probability) and the estimates provided by the NFs. The results indicate a minimal difference, further validating the effectiveness of the NFs in estimating the future state density (goal-reaching probability).

where $f(s, a)$ denotes the deterministic transition function. The future discounted state distribution is then given by:

$$
\begin{aligned}
P &= (1 - \gamma) \left[ T_0 + \gamma T_0 T + \gamma^2 T_0 T^2 + \gamma^3 T_0 T^3 + \cdots \right] \\
&= (1 - \gamma) T_0 \left[ I + \gamma T + \gamma^2 T^2 + \gamma^3 T^3 + \cdots \right] \\
&= (1 - \gamma) T_0 \left( I - \gamma T \right)^{-1}
\end{aligned}
$$

The tensor-matrix product $T_0 T$ is equivalent to `einsum`('ijk,kh $\to$ ijh', $T_0$, $T$). We use the forward KL divergence for estimating the error in our estimate, $D_{\mathrm{KL}}(P \| Q)$, where $Q$ is the tensor of predictions:

$$
Q \in \mathbb{R}^{25 \times 4 \times 25} : \quad Q[s, a, g] = q(g \mid s, a).
$$

Following the configuration outlined in Eysenbach et al. (2021), we compare the accuracy of the future discounted state distribution under against C-Learning and $Q$-learning:

**On-policy Setting** Figure 17 presents the results of our evaluation comparing CVAE, C-learning, CRL and NFs on the above modified "continuous GridWorld" environment under the on-policy setting. In this scenario, CVAE demonstrates higher error compared to C-learning, while NFs achieves the best performance. This highlights the accuracy of NFs in estimating the discounted state occupancy measure. This experiment aims to answer whether NFs solve the future state density estimation problem.

### G.5. Can Expectile Regression Effectively Capture Maximum $Q$-values in Practice?

We empirically validate that expectile regression converges to in-distribution maximum $Q$-values in a controlled GridWorld setting, supporting our theoretical analysis in Theorem 3.1.

**Metrics.** We use coefficient of determination ($R^2$) measuring explained variance, and Mean Absolute Error (MAE) quantifying prediction deviation:

$$
R^2 = 1 - \frac{\sum (y_{\text{true}} - y_{\text{pred}})^2}{\sum (y_{\text{true}} - \bar{y}_{\text{true}})^2}, \quad \text{MAE} = \frac{1}{n} \sum_{i=1}^{n} |y_{\text{true},i} - y_{\text{pred},i}|. \tag{59}
$$

**Results.** As shown in Figures 18 and 19, the results strongly support our theoretical analysis:

1. Standard MSE ($\tau = 0.5$) learns the mean rather than maximum, yielding $R^2 = 0.781$;

2. Performance improves monotonically with $\tau$: $R^2$ increases from $0.781$ to $0.995$ as $\tau$ goes from $0.5$ to $0.99$;

3. At $\tau = 0.99$, predicted values closely match ground-truth $Q_{\max}^{\beta}$ with $R^2 = 0.995$ and MAE$= 0.0017$.

**Implications.** These results validate that expectile regression effectively captures maximum in-distribution $Q$-values, which is essential for **QHyer**'s trajectory stitching capability. The convergence aligns with Theorem 3.1: as $\tau \to 1$, the approximation error $\epsilon_\tau \to 0$ and $\hat{Q}^\tau \to Q^\star$. Specifically, our theoretical bound predicts:

$$\epsilon_\tau \leq \frac{(1 - \tau)(Q^\star - Q_{\min})}{\tau \cdot \tilde{c}/2 + (1 - \tau)(1 - \tilde{c}/2)}, \tag{60}$$

which decreases as $\tau$ increases, consistent with the monotonic improvement observed in Figure 19.

However, excessively large $\tau$ (e.g., $0.999$) may cause overfitting to outliers due to focusing on too few high-value samples, leading to increased variance. In practice, $\tau \in [0.9, 0.95]$ balances accuracy and training stability, as validated in our ablation studies (Section 4.3).

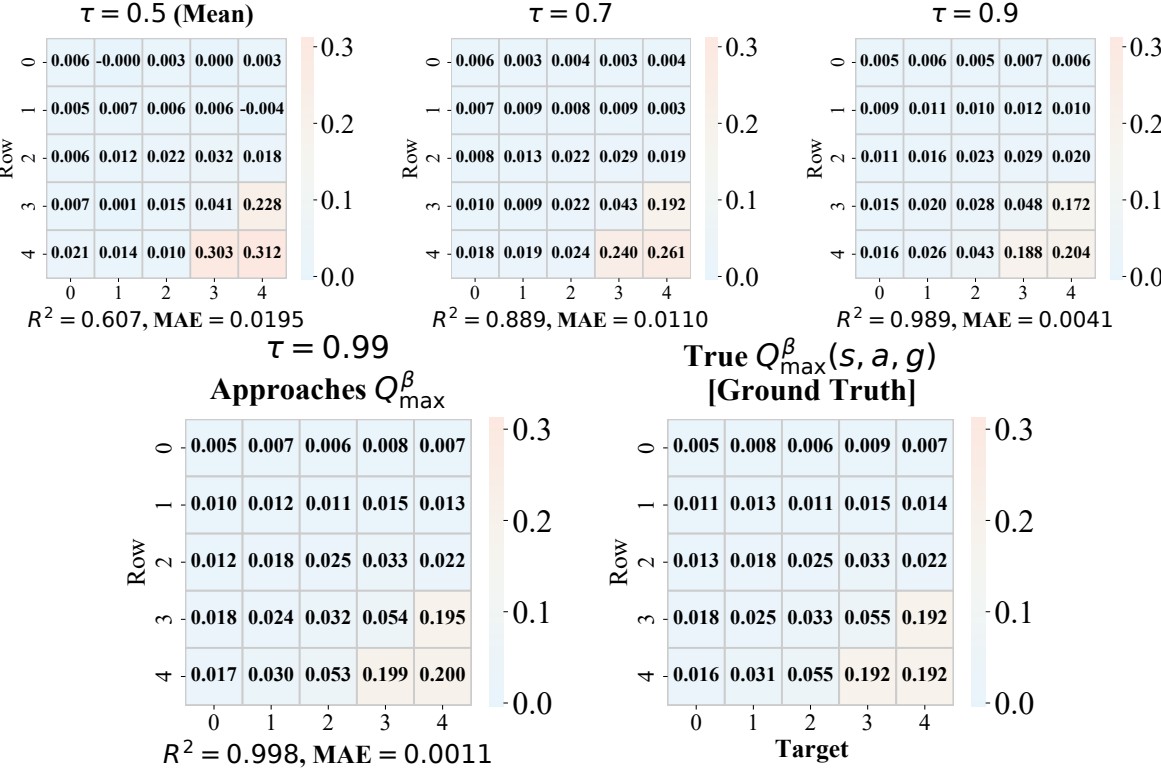

*Figure 18.* Visualization comparing predicted maximum $Q$-values from expectile regression with different $\tau$ values against ground-truth maximum $Q$-values ($Q^\star$) in the GridWorld environment. As $\tau$ increases from $0.5$ to $0.99$, the predictions converge toward the diagonal line (perfect prediction), validating Theorem 3.1.

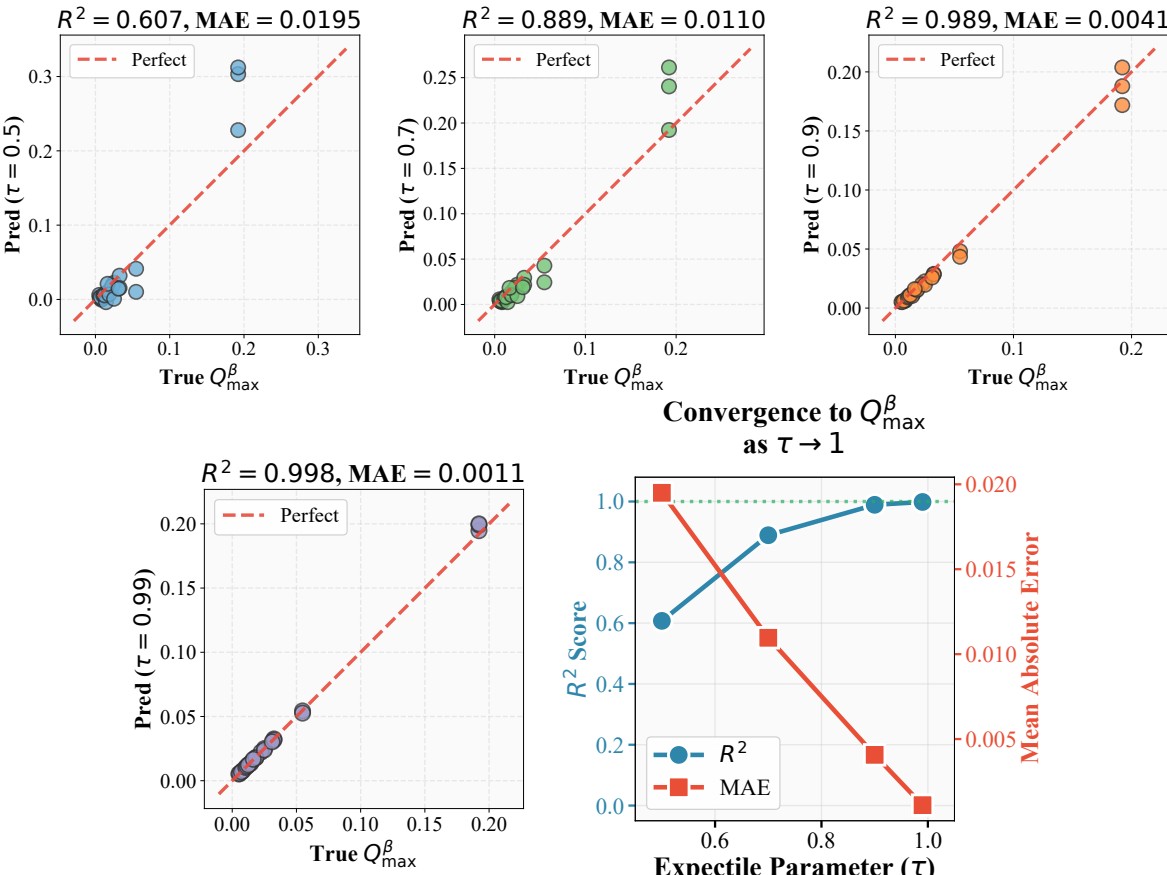

*Figure 19.* Curves of $R^2$ and MAE metrics as a function of the expectile parameter $\tau$. $R^2$ increases monotonically from 0.781 ($\tau = 0.5$) to 0.995 ($\tau = 0.99$), while MAE decreases correspondingly. This empirical trend confirms that expectile regression with high $\tau$ effectively approximates the in-distribution optimal Q-value $Q^\star$, consistent with the theoretical bound in Theorem 3.1.

### G.6. Comparison with Recent Offline RL Methods Adapted to GCRL

To strengthen baseline coverage, we adapt four recent offline RL methods to offline GCRL by attaching HER goal relabeling and following the OGBench evaluation protocol. These are Transitive RL (Park et al., 2026b), SHARSA (Park et al., 2026a), DEAS (Kim et al., 2026), and QCFQL (Li et al., 2026). Results are averaged over 8 seeds at 1M training steps.

*Table 11.* **Recent offline RL methods adapted to offline GCRL on OGBench `play` datasets.** Mean success rate (%) over 8 seeds. **Orange** = best, underline = second best.

| Environment | GC-TrL | GC-SHARSA | GC-DEAS | GC-QCFQL | QHyer |
|---|---|---|---|---|---|
| `cube-single-play` | $46.2_{\pm0.8}$ | $30.7_{\pm3.8}$ | $44.1_{\pm2.6}$ | $38.4_{\pm1.5}$ | $84_{\pm4}$ |
| `cube-double-play` | $1.6_{\pm0.8}$ | $49.3_{\pm4.7}$ | $10.7_{\pm3.0}$ | $5.1_{\pm0.0}$ | $56_{\pm2}$ |
| `cube-triple-play` | $0.8_{\pm0.6}$ | $36.8_{\pm4.0}$ | $18.1_{\pm2.3}$ | $1.4_{\pm0.7}$ | $10_{\pm5}$ |
| `cube-quadruple-play` | $0.0_{\pm0.0}$ | $2.3_{\pm0.3}$ | $0.1_{\pm0.1}$ | $0.0_{\pm0.0}$ | $2_{\pm1}$ |
| `scene-play` | $27.7_{\pm6.2}$ | $44.0_{\pm9.1}$ | $48.7_{\pm3.4}$ | $36.9_{\pm4.5}$ | $53_{\pm2}$ |
| `puzzle-3x3-play` | $7.6_{\pm1.6}$ | $35.6_{\pm2.5}$ | $32.7_{\pm11.0}$ | $14.9_{\pm9.6}$ | $92_{\pm2}$ |
| `puzzle-4x4-play` | $2.3_{\pm0.7}$ | $32.4_{\pm9.7}$ | $26.7_{\pm6.7}$ | $0.3_{\pm0.7}$ | $28_{\pm5}$ |
| `puzzle-4x5-play` | $2.0_{\pm1.5}$ | $15.1_{\pm3.1}$ | $16.1_{\pm1.3}$ | $9.7_{\pm3.1}$ | $31_{\pm1}$ |
| `puzzle-4x6-play` | $1.6_{\pm0.5}$ | $12.1_{\pm2.4}$ | $17.9_{\pm4.5}$ | $8.3_{\pm5.5}$ | $18_{\pm2}$ |
| *Average* | *10.0* | *28.7* | *23.9* | *12.8* | *41.6* |

**Interpretation.** Three essential reasons explain the gap. First, none of these methods were originally validated under offline GCRL with sparse binary rewards at standard OGBench scale. TRL and SHARSA rely on oracle goals or the large-data regime, DEAS targets semi-sparse single-task settings, and QCFQL's strongest numbers come from offline-to-online training. When forced into the pure offline, sparse-binary, multi-goal regime, their value targets and exploration mechanisms become mis-specified. Second, there are structural mismatches. TRL's triangle inequality on temporal distance holds for continuous navigation but breaks under manipulation's discrete contact-mode transitions, which is why TRL drops from $46.2$ on `cube-single` to $1.6$ on `cube-double`. Third, SHARSA must predict subgoals in the full multi-object pose space, which is far harder than 2D navigation waypoints, and DEAS and QCFQL execute fixed-length open-loop action chunks, so early errors compound and the fixed chunk length cannot align with variable-duration manipulation primitives. SHARSA and DEAS nonetheless remain nontrivially competitive on the hardest long-horizon tasks, suggesting that action chunking and temporal abstraction are complementary to our contributions.

### G.7. Comparison with Graph-based Stitching (GAS)

We also compare against GAS (Baek et al., 2025), a graph-based offline GCRL stitching method.

*Table 12.* **Comparison with GAS on navigation and visual manipulation.** Mean success rate (%) over 5 seeds. **Orange** = best.

| Environment | GAS | QHyer |
|---|---|---|
| `antmaze-giant-stitch` (navigation) | $88_{\pm4}$ | $70_{\pm2}$ |
| `visual-scene-play` (manipulation) | $54_{\pm6}$ | $96_{\pm1}$ |

**Interpretation.** GAS and **QHyer** occupy complementary regimes, and the reason is structural rather than a matter of tuning. On `antmaze-giant-stitch`, GAS replaces high-level policy learning with Dijkstra shortest-path search over a precomputed temporal-distance graph, which directly exploits the metric structure of continuous navigation. **QHyer**'s flat sequence model cannot match this advantage on pure navigation. On `visual-scene-play`, the tables turn. GAS's graph construction is bottlenecked by its ability to learn high-dimensional representations, whereas **QHyer**'s end-to-end sequence modeling with content-adaptive memory benefits directly from pixel inputs, producing roughly a 42-point improvement and nearly doubling the previous OGBench best. We therefore view the two methods as complementary tools rather than directly competing baselines.

