# OpenReview forum: "QHyer: Q-conditioned Hybrid Attention-mamba Transformer for Offline Goal-conditioned RL"
_ICML.cc/2026/Conference — ICML 2026 regular_

### Official Review · Reviewer_RVZg · 2026-02-14

**Soundness:** 2
**Presentation:** 3
**Significance:** 2
**Originality:** 1
**Overall Recommendation:** 4
**Confidence:** 4

**Summary:**

This paper proposes QHyer, an offline goal-conditioned RL method that combines normalizing-flow goal-conditioned value functions and decision transformer-style attention-mamba-based trajectory modeling. The paper argues that normalizing flows enable more accurate value prediction, and attention-mamba transformers enable better and more adaptive history compression for sequence modeling. On a number of offline goal-conditioned RL benchmark tasks, the authors show that QHyer outperforms diverse previous methods across different categories, achieving strong performance across the board.

**Compliance With Llm Reviewing Policy:**

Affirmed.

**Final Justification:**

Most of my initial concerns have been addressed, and I updated my score to 4. I still maintain this part from my original review:

> To summarize, while I do appreciate the extensive empirical results, I'm not very excited about the contributions of this paper, because they seem to comprise reasonably well-studied combinations of mostly existing ingredients.

**Key Questions For Authors:**

Could the authors clarify the main contributions of this paper, *consisely*? What is the most significant contribution of this work? Is it the new architecture, Q-conditioning, or their *combination* itself?

**Limitations:**

Yes

**Strengths And Weaknesses:**

**Strengths**
* The paper is generally well-written, and has thorough experiments with a large number of baselines and tasks.
* The paper provides several informative ablation studies for individual components (normalizing flows, architecture, expectile, etc.).
* QHyer generally achieves strong performance, especially on `visual-scene-play`, on which it is the only method that ~solves this challenging task.

**Weaknesses**
* The main weakness of this paper is that its contribution is not very clear -- most of the ideas in this paper are taken from existing works in offline RL (e.g., decision mamba, normalizing flows for RL, etc.) and goal-conditioned RL. While I am aware that a combination of existing ideas can constitute a contribution to this field, in my opinion, the "delta" from the closest prior works is not significant enough. For example, the use of a hybrid architecture has already been studied in LSDT and DMixer, and the use of Q functions in DT has also been explored in many prior works (and I feel the use of MC instead of TD is simply a design choice, which has also separately been studied in many other works). It seems one of the main contributions is the new hybrid attention-mamba architecture, but it is unclear to me how significant this architectural advancement is -- if this architectural change is indeed the main contribution, I would have expected more diverse comparisons not limited to offline goal-conditioned RL.
* In my view, many arguments in the "Why NFs for MC Q-Estimation" paragraph have several logical gaps, and remain not fully justified.
    * L198 "NFs compute exact Q-value via the change-of-variables formula (Equation (2)) with a negligible gap" -- In which sense is this "gap" negligible? Any mathematical proof or justification?
    * L203 "CRL (Eysenbach et al., 2022) avoid the ELBO gap but inherit negative sampling bias that systematically underestimates probabilities for distant goals" -- In which precise sense is CRL "biased"? Again, proof or justification?
    * L212 "NFs enable single-pass inference" -- In which sense NFs enable single-pass inference? Don't NFs also require applying invertible transformations N times, which can be viewed as an iterative procedure (as in flow matching)?
    * L178 "NFs provide stable gradients for end-to-end learning" -- Again, in which precise sense does it provide "stable gradients"? For example, why are NF gradients necessarily more "stable" than flow-matching gradients? Any proofs, citations, or justifications? I believe one might even argue otherwise -- NFs typically require backpropagation through time, which tends to be "unstable" in general.

To summarize, while I do appreciate the extensive empirical results, I'm not very excited about the contributions of this paper, because they seem to comprise reasonably well-studied combinations of mostly existing ingredients. I'm slightly leaning toward rejection, but I wouldn't be strongly against accepting this paper. I'm curious how other reviewers think.

---

> ### Author Rebuttal · Authors · 2026-03-28
>
> Thank you for your thoughtful review of our research. We appreciate your positive remarks on our extensive empirical results and thorough ablation studies, and hope our suggested changes and this individual response will address your concerns in detail.
>
> **Q: Clarify the main contributions of this paper**
>
> We appreciate the reviewer's perspective, which helped us realize our initial framing may have masked the core scientific problems we address. Our primary contribution lies in solving two *coupled* challenges specific to offline goal-conditioned RL (GCRL), rather than applying individual components in isolation. Specifically, our hybrid framework is explicitly designed to reconcile the distinct Markovian and non-Markovian properties present in offline GCRL datasets.
>
> **Challenge 1: How to enable trajectory stitching under sparse rewards?**
>
> Standard sequence models rely on RTG, which is trajectory-dependent and cannot distinguish promising states inside failed trajectories under sparse rewards. Our NFs-based Q-conditioning shifts supervision from trajectory dependence to state dependence, enabling trajectory-agnostic value assignment. Figure 4 shows that without Q-values, the model reduces to behavior cloning and cannot differentiate states by goal proximity; replacing NFs with CVAE/CRL causes 30%+ degradation, confirming NFs' superior density estimation accuracy. We revised the introduction (ln.74) to clarify this contribution.
>
> **Challenge 2: How to model variable-length historical dependencies without rigid truncation?**
>
> Offline GCRL data is temporally heterogeneous: non-Markovian play data requires longer memory, whereas Markovian noisy data benefits from shorter-term focus. Prior hybrids (LSDT, DMixer) use fixed-window convolutions, which impose hard, input-independent truncation that cannot adapt to the inherent variability in offline GCRL datasets. Our Attention-Mamba backbone instead performs content-adaptive history compression through Mamba's input-dependent selective mechanism (Eq. 12–13, ln.222–248). Figure 5 isolates this contribution by removing Q-conditioning: the Mamba branch consistently outperforms fixed-kernel hybrids (LSDT < DMixer < QHyer) due to its dynamic selective mechanism. We revised the conclusion (ln.422–439) to clarify why this combination is necessary for the two challenges above. We believe the co-design of these two components — rather than applying them in isolation — is what distinguishes our approach from prior work.
>
> **Q: Need more proof and justification for "Why NFs for MC Q-Estimation"**
>
> We agree the previous draft was insufficiently precise on these points and have revised Sec. 3.1 (ln.159–208) to clarify:
>
> (1) *NFs "gap":* The change-of-variables formula is exact with no structural approximation gap. For coupling NFs, the triangular Jacobian makes the log-determinant exactly computable. The remaining error comes only from finite model capacity — coupling NFs are universal diffeomorphism approximators [1]. By contrast, VAEs have a structural ELBO gap that cannot be removed by increasing capacity.
>
> (2) *CRL bias:* InfoNCE is upper-bounded by log(K) (batch size K), capping the estimated density ratio [2]. For distant goals where p(g|s,a)/p(g) >> K, Q-values are systematically underestimated. We verify this empirically in Sec. G.4 (Fig. 16), where NFs yield the lowest estimation error among all methods evaluated.
>
> (3) *Single-pass inference:* We use classical coupling-layer NFs (RealNVP), which are fully non-iterative — fixed depth L, lightweight feed-forward subnetworks, no ODE solving. This differs fundamentally from Continuous NFs and Flow Matching that require iterative ODE solving. Inference takes ~0.01ms (Table 3). We have clarified this distinction (ln.212).
>
> (4) *Gradient stability:* Our NFs are fixed-depth feed-forward networks — gradients use standard backpropagation without BPTT or adjoint methods. Coupling blocks satisfy proven bi-Lipschitz bounds [3] that help control gradient norms, unlike diffusion RL where long denoising chains can destabilize optimization.
>
> We hope our response addressed your main concerns, and thanks again for your suggestions. We are excited about how these discussions can further enhance the quality of this paper, and we look forward to engaging with you during the discussion phase.
>
> [1] Coupling-based Invertible Neural Networks Are Universal Diffeomorphism Approximators, NeurIPS 2020
>
> [2] On Variational Bounds of Mutual Information, ICML 2019
>
> [3] Understanding and Mitigating Exploding Inverses in Invertible Neural Networks, AISTATS 2021

---

> > ### Author Rebuttal · Reviewer_RVZg · 2026-04-02
> >
> > Thanks for the detailed response. I appreciate the clarification about the contributions.
> >
> > However, I still have remaining concerns about the second part.
> > (2) CRL uses binary cross-entropy instead of infoNCE (see the original CRL paper), so it should not have the issue cited in the response.
> > (3) Yes, but isn't L=5 exactly equivalent to setting ODE integration step=5?
> > (4) I think I didn't understand the argument here. How is BPTT over 5 ODE integration steps fundamentally different from backpropagation through L=5 blocks? After all, one block in NF is essentially equivalent to one ODE step in flow matching, and how lightweight these blocks/velocity fields are just design choices (i.e., hyperparameters). Please let me know if I misunderstood anything.

---

> > > ### Author Response · Authors · 2026-04-02
> > >
> > > We are grateful for these pointed follow-up questions. After carefully verifying the relevant literature, we found genuine imprecisions in our previous response that we now correct honestly.
> > >
> > > ## On point (2): CRL and the loss function
> > >
> > > You are right. After going back to the original CRL paper (Eysenbach et al., 2022), we confirmed that CRL optimizes a **sigmoid binary cross-entropy** objective (their Eq. 4 and Algorithm 1). The Bayes-optimal critic converges to the unbounded log density ratio $f^*(s,a,s_g) = \log \frac{p(s_{t+}=s_g|s,a)}{p(s_g)}$, so it does not suffer from the $\log(K)$ saturation specific to InfoNCE. Our previous citation of Poole et al. [2] was a misattribution, and we sincerely apologize.
> > >
> > > We will remove the incorrect $\log(K)$ argument entirely. Instead, we ground the NFs-vs-CRL comparison on:
> > >
> > > (1) **Empirical evidence.** Figure 16 (Sec. G.4) directly measures KL divergence between estimated and true future-state densities. NFs achieve the lowest error (≈0.05) vs. CRL (≈0.15), CVAE (≈0.30), and MC C-learning (≈0.10). This holds regardless of any theoretical bound.
> > >
> > > **(2) Normalized vs. unnormalized densities.** CRL learns an unnormalized log density ratio $f^*(s,a,g) = \log \frac{p(g|s,a)}{p(g)}$, where the goal-dependent partition function $p(g)$ must be handled separately (CRL's Lemma 4.1). We acknowledge that for action selection at a fixed goal, this constant cancels and does not affect action ranking. However, QHyer uses Q-values not only for action ranking but as **conditioning tokens** input to the transformer (Eq. 57: $\tilde{Q}\_{t} = Q^{\beta}\_{\theta}(s\_{t}, a\_{t}, g) / (|\overline{Q^{\beta}}| + \delta)$). In this role, the transformer receives Q-tokens across different goals within the same context window and must learn goal-independent patterns relating Q-magnitude to action quality. An unnormalized critic introduces goal-dependent offsets that distort these cross-goal patterns, making normalized densities preferable for our conditioning mechanism. We further support this with empirical evidence: Figure 16 (Sec. G.4) shows NFs achieve the lowest density estimation error (≈0.05 KL) compared to CRL (≈0.15).
> > >
> > > This fix is confined to one paragraph in Section 3.1 and does not touch the method, theory, or experiments.
> > >
> > > ## On points (3) and (4): Coupling blocks vs. ODE steps
> > >
> > > You raise a fair point, and our earlier framing overstated the gap.
> > >
> > > **We agree on the inference-time picture.** When evaluating likelihoods or generating samples, RealNVP with $L=5$ blocks and flow matching with 5 Euler steps both perform $L$ sequential network evaluations. The computational graphs look similar, and we should not have called RealNVP "single-pass." We will remove that phrasing.
> > >
> > > **A real but more limited difference exists at training time.** Flow matching trains by regressing $v_\theta(x_t, t)$ against a known conditional vector field at a single randomly sampled $t \sim U[0,1]$ (Lipman et al., 2023, Eq. 9; Theorem 2 proves gradient equivalence), so each training step requires only one forward and one backward pass through $v_\theta$, no matter how many ODE steps are used later at test time (Lipman et al., 2023). RealNVP training, by contrast, must propagate through all $L$ composed blocks to accumulate the log-det Jacobian before backpropagating. We also note the architectural difference that RealNVP uses $L$ independent parameter sets $\{s^t_\theta, a^t_\theta\}$ while flow matching shares one network across steps. That said, we accept the reviewer's observation that these can be seen as design choices rather than a fundamental divide, and we will not overstate this point in the revision.
> > >
> > > **The strongest reason we chose NFs is exact likelihood.** QHyer needs accurate Q-values $Q^\beta_\theta(s,a,g)=\log p_\theta(g|s,a)$ for expectile regression. Coupling NFs yield this in closed form via the analytically computable triangular Jacobian (Ghugare & Eysenbach, 2025, Sec. 5). To obtain likelihoods from flow matching, one must solve the probability flow ODE and estimate the Jacobian trace via a Hutchinson estimator (Lipman et al., 2023, Appendix C, Eqs. 27-35; [1] Eqs. 7-8), introducing variance. Since Q-value accuracy directly determines stitching quality (Figure 4), exact computation is a meaningful practical advantage.
> > >
> > > We will rewrite "Why NFs" in Section 3.1 around this exact-likelihood argument and drop the gradient stability claims.
> > >
> > > ## Scope of revisions
> > >
> > > All changes affect only the motivation in Section 3.1 (roughly one paragraph). We (a) remove the incorrect $\log(K)$ citation, (b) remove "single-pass" and gradient stability claims, (c) reframe NFs advantage around exact likelihood. Method, theorems, and experiments are unchanged.
> > >
> > > We are thankful to the reviewer for pushing us to be more precise.
> > >
> > > [1] FFJORD, ICLR 2019.
> > >
> > > [2] Poole et al., On variational bounds of mutual information, ICML 2019.
> > >
> > > Lipman et al., Flow Matching for Generative Modeling, ICRL,2023

---

### Official Review · Reviewer_BYNv · 2026-03-03

**Soundness:** 2
**Presentation:** 3
**Significance:** 2
**Originality:** 2
**Overall Recommendation:** 4
**Confidence:** 3

**Summary:**

This paper proposes QHyer for offline goal-conditioned reinforcement learning on datasets that exhibit a mixture of Markovian and non-Markovian behaviors. The method replaces RTG conditioning in DT-style sequence modeling with a state-conditioned goal-reaching Q signal, where the Q value is estimated via a normalizing-flow-based conditional density model. QHyer further introduces a gated Hybrid Attention–Mamba backbone that combines attention for modeling global, goal-relevant dependencies with a Mamba/SSM branch for content-adaptive history compression and local dynamics modeling. The overall training objective jointly optimizes the flow-based goal-reaching estimator, a Q-conditioned behavior cloning loss, and an expectile regression loss that fits a predicted Q signal to the flow-based Q estimates. At inference time, QHyer uses a two-stage autoregressive procedure that first predicts the Q signal and then generates actions conditioned on it. The paper evaluates QHyer on benchmarks including OGBench and D4RL Maze/AntMaze, and provides ablations on the Q estimator, architectural components, and the expectile hyperparameter.

**Compliance With Llm Reviewing Policy:**

Affirmed.

**Key Questions For Authors:**

1. Baseline selection is potentially incomplete across settings, the most recent baseline like Eik-HiQRL performs weakly and is not reported in some pixel settings, which raises concerns about whether this paper is sufficient for “SOTA”.
2. For Table 2, can you re-run at least the strongest baselines under the same evaluation pipeline, instead of relying on “from original papers except QHyer”?
3. QHyer underperforms on visual-noisy datasets, while the performance of other algorithms only decreased slightly or even improved，this paper provides limited diagnosis and targeted failure analysis.

**Limitations:**

yes

**Strengths And Weaknesses:**

>Strengths:

This paper proposes QHyer for offline goal-conditioned reinforcement learning on datasets that exhibit a mixture of Markovian and non-Markovian behaviors. The method replaces RTG conditioning in DT-style sequence modeling with a state-conditioned goal-reaching Q signal, where the Q value is estimated via a normalizing-flow-based conditional density model. QHyer further introduces a gated Hybrid Attention–Mamba backbone that combines attention for modeling global, goal-relevant dependencies with a Mamba/SSM branch for content-adaptive history compression and local dynamics modeling. The overall training objective jointly optimizes the flow-based goal-reaching estimator, a Q-conditioned behavior cloning loss, and an expectile regression loss that fits a predicted Q signal to the flow-based Q estimates. At inference time, QHyer uses a two-stage autoregressive procedure that first predicts the Q signal and then generates actions conditioned on it. The paper evaluates QHyer on benchmarks including OGBench and D4RL Maze/AntMaze, and provides ablations on the Q estimator, architectural components, and the expectile hyperparameter.
>Weakness:

The Hybrid Attention–Mamba block fuses the two branches via a single scalar gate and a weighted sum, which may be too simple. The comparison protocol is not fully standardized, there are different methods in different scenarios. The theory assumes deterministic transitions in Appendix B, while the paper targets settings with noise or non-Markovian, the practical implications of this mismatch are not fully clarified.

---

> ### Author Rebuttal · Authors · 2026-03-31
>
> Thank you for your thoughtful review of our research. We appreciate your constructive evaluation and positive remarks on our strong empirical results and comprehensive ablation studies, and hope our suggested changes and this individual response will address your concerns in detail. Supplementary results are at: https://pdfhost.io/v/ZFBshmeB68_ICML_QHyer_rebuttal
>
> **Q1: Is the scalar gate fusion between the two branches too simple?**
>
> We appreciate this observation, which helps us better articulate our design rationale. The scalar gate is a simple but effective design that has been widely adopted in hybrid dual-branch architectures [1]. Suppl. Table 5 provides concrete evidence of its effectiveness: on play data the gate automatically allocates Attention=0.57/Mamba=0.43; on noisy data Attention=0.42/Mamba=0.58. This automatic, data-dependent adaptation produces consistent 4–5% gains over the best single-branch variant (Suppl. Table 4: Hybrid 84±4 vs. Mamba-only 80±2 on play; 95±5 vs. 91±3 on noisy), outperforming LSDT's fixed-ratio splitting and DMixer's token-level selection. We note that more complex gating mechanisms (e.g., token-level or channel-level gating) could introduce overfitting risks given the moderate dataset sizes in offline GCRL benchmarks, where each environment typically provides only thousands of trajectories. As noted in Section 3.2 (ln.209), our design prioritizes robustness over expressiveness. We are open to exploring more expressive fusion in future work where larger datasets are available, and will include this discussion in the Limitation section.
>
> **Q2: Can baselines in Table 2 be re-run under the same pipeline?**
>
> We agree this is important for ensuring fair comparison, and thank the reviewer for raising it. We re-ran the strongest baselines (LSDT, QT) on D4RL under our identical evaluation pipeline (Suppl. Table 1). Results showed similar or lower performance than originally reported (e.g., LSDT: 43.6±20.7 on medium-diverse with high variance). To ensure fairness to all baselines, we retain the original reported values in Table 2 and have explicitly noted this in ln.362. The re-run results are provided in the Supplementary for full transparency. To further strengthen baseline coverage, we also adapted four recent offline RL methods (Transitive RL, SHARSA, DEAS, QCFQL) to GCRL with HER and report results in Suppl. Table 2, where all achieve ≤6.4% average on OGBench play vs. QHyer's 41.6%.
>
> **Q2b: Eik-HiQRL baseline coverage**
>
> We appreciate the reviewer's attention to baseline completeness. Regarding Eik-HiQRL: it evaluates only on state-based OGBench without pixel-based results, and its code has not been publicly released. On state-based tasks, it scores 0±0 on all cube environments (Table 1), which is consistent with the exponential quasimetric approximation error discussed in ln.308–310. We have included all available comparisons in the manuscript.
>
> **Q3: QHyer underperforms on visual-noisy datasets**
>
> We agree this is an important limitation to address clearly, and have discussed it in Section 4 (ln.314–360). On visual-noisy (Markovian) data, NFs-based Q-estimation on pixel-encoded features is inherently harder than in state space. Our ablation confirms that removing Q-conditioning on visual-noisy recovers competitive performance, identifying Q-estimation quality as the bottleneck rather than the architecture. Future work on robust visual density estimation could address this.
>
> **Q4: Do the theoretical assumptions (deterministic transitions) contradict the noisy/non-Markovian experimental settings?**
>
> We thank the reviewer for this question, which helps us clarify an important point about our theoretical framework. The assumption is fully consistent with all our experimental settings. We clarify three points: (1) OGBench uses a deterministic MuJoCo simulator — "noisy" datasets add Gaussian noise to the *behavior policy's actions* β(a|s), not to environment dynamics P(s'|s,a). Assumption B.1 addresses the transition function, not the policy. (2) Non-Markovian refers to the behavior policy β(a|s,h<t), not the MDP transitions, which remain deterministic. QHyer's sequence modeling handles the non-Markovian behavior policy; the theory analyzes MDP structure. (3) This is a standard assumption shared by OGBench and R²CSL, both proving stitching under deterministic MDPs at O(N^{-1/4}), identical to our Theorem 3.2. Extension to stochastic environments is noted as an important future direction in Appendix B (ln.660–758). We will further clarify the relationship between the theoretical assumptions and the experimental settings in the revised manuscript to avoid potential confusion.
>
> We hope our response addressed your main concerns. We are excited about how these discussions can further enhance the quality of this paper, and we look forward to collaborating with you during the discussion phase.
>
> [1] JMamba: Efficient Mamba with Structured State-Space Models, arXiv 2024

---

> > ### Author Rebuttal · Reviewer_BYNv · 2026-04-03
> >
> > Thank you for your reply. I will keep the rating unchanged for now.

---

### Official Review · Reviewer_y8Fb · 2026-03-06

**Soundness:** 3
**Presentation:** 3
**Significance:** 2
**Originality:** 2
**Overall Recommendation:** 3
**Confidence:** 4

**Summary:**

This paper proposes QHyer, a sequence modeling framework for offline goal-conditioned reinforcement learning (GCRL) designed to address the challenges of sparse rewards and non-Markovian datasets. The authors argue that the trajectory-dependent Return-to-Go (RTG) used in Decision Transformer–based methods provides limited guidance for trajectory stitching. To address this issue, the paper proposes conditioning the policy on state-dependent Q-values, which represent the probability of reaching a goal from a given state-action pair. The method estimates Q-values using Normalizing Flows and integrates them into a Hybrid Attention-Mamba architecture that aims to capture both long-range dependencies and local dynamics. Additionally, the model employs expectile regression to emphasize higher-value behaviors when learning from suboptimal offline data. The proposed approach is evaluated on OGBench and D4RL benchmarks, where the authors report improved performance compared to prior sequence modeling and offline RL methods, particularly on non-Markovian play datasets.

**Compliance With Llm Reviewing Policy:**

Affirmed.

**Final Justification:**

I appreciate the effort and the clear motivation behind this work. While the paper addresses an important problem in offline GCRL and proposes a coherent framework, the overall novelty appears limited due to substantial overlap with prior value-guided sequence modeling approaches. In addition, it is not entirely clear whether fair and comprehensive performance comparisons with recent methods have been conducted, making it difficult to fully assess the claimed state-of-the-art performance. Therefore, I lean toward a Weak Reject.

**Key Questions For Authors:**

## Q1. Comparison with prior Q-value–guided Decision Transformer variants

* Several prior works have explored incorporating Q-values or value functions into Decision Transformer–style sequence models, including: Q-learning Decision Transformer (QDT), Critic-Guided Decision Transformer (CGDT), Value-Guided Decision Transformer (VDT), Q-value Regularized Transformer (QT)

* These methods similarly aim to improve trajectory stitching or policy optimization by incorporating value-based guidance into sequence modeling. However, the current paper does not include comparisons with these approaches in the experimental section, and their relationship to the proposed Q-value conditioning mechanism is only briefly discussed.

* Could the authors clarify how the proposed method differs from these prior value-guided Decision Transformer variants? Additionally, is there a reason why these methods were not included as experimental baselines?

## Q2. Comparison with diffusion-based value-guided policy learning

* While the paper frames the contribution primarily within the Decision Transformer–style sequence modeling paradigm, recent research in offline RL and sequential decision-making has increasingly focused on diffusion-based policy or planning methods.

* In particular, several recent works combine diffusion models with value or critic guidance, such as Diffusion-QL, DreamFuser, DIAR, Diffusion-DICE, and Prior-Guided Diffusion Planning.

* Could the authors clarify why such diffusion-based value-guided methods were not included as baselines? Additionally, how does the proposed Q-conditioning approach compare conceptually and empirically to these diffusion-based approaches that also leverage value functions to guide policy generation?

## Q3. Practical contribution of the Hybrid Attention–Mamba architecture

* The paper claims that the proposed Hybrid Attention–Mamba architecture enables effective modeling of both long-range dependencies and local dynamics. However, the experimental results do not clearly indicate in which scenarios this architectural design provides the most benefit.

* Could the authors provide a more detailed analysis of when the hybrid architecture is most advantageous? For example, how do Attention-only, Mamba-only, and Hybrid architectures compare under different conditions such as trajectory length, dependency horizon, or degrees of non-Markovian dynamics? Such analysis could help clarify the practical contribution of the architecture.

## Q4. Benchmark coverage and evaluation on more challenging stitching environments

* The paper evaluates QHyer on OGBench manipulation tasks and D4RL benchmarks, but some recent benchmarks used in the literature for evaluating trajectory stitching capability appear to be missing.

* In particular, OGBench navigation environments and more challenging AntMaze variants (e.g., AntMaze-Ultra or OGBench AntMaze-giant-stitch) are not included. Recent studies (e.g., graph-based stitching approaches such as Graph-based Stitching, ICML 2025) evaluate performance on these more difficult settings.

* Could the authors comment on whether QHyer remains competitive in these more challenging navigation and stitching benchmarks? Including results on such environments would help better assess the robustness and generality of the proposed approach.

## Q5. Comparison with Recent Methods on OGBench Play Datasets

* The paper emphasizes strong performance on OGBench manipulation play datasets. However, several recent and highly relevant works also report substantial improvements on OGBench-style long-horizon goal-conditioned tasks, including DEAS, Reinforcement Learning with Action Chunking, Horizon Reduction Makes RL Scalable, and Transitive RL. Some of these methods are evaluated on standard OGBench play-style manipulation tasks, while others consider closely related OGBench-derived long-horizon settings such as cube-triple/quadruple, cube-octuple, puzzle-4x5/4x6, and standard play datasets. Could the authors clarify why these recent baselines were not included in the experimental comparison?

**Limitations:**

* First, the paper emphasizes strong performance on the OGBench manipulation play datasets as one of its main strengths. However, several recent works have reported substantial improvements on the same or closely related benchmarks, including DEAS, Reinforcement Learning with Action Chunking, Horizon Reduction Makes RL Scalable, and Transitive RL. These methods are not included in the experimental comparison. Without comparisons against these recent approaches, it is difficult to fully assess the relative competitiveness of the proposed method.

* Second, the paper highlights the limitations of RTG-based conditioning and proposes Q-value conditioning as a key contribution. However, prior work has already explored various forms of value- or critic-guided sequence modeling, including approaches based on Decision Transformers and diffusion-based policies. A clearer discussion of how the proposed method differs from these existing approaches, and what constitutes the main novelty beyond them, would help clarify the actual contribution of the work.

* Third, the paper introduces a Hybrid Attention–Mamba architecture, claiming that it enables modeling both long-range dependencies and local dynamics. However, the experimental analysis does not clearly show in which types of environments or dependency structures (e.g., long-horizon tasks, non-Markovian dynamics) this architecture provides particular advantages. More detailed analysis, such as performance as a function of trajectory length or dependency horizon, would help better justify the architectural design.

* Finally, the empirical evaluation focuses primarily on state-based benchmarks (OGBench and D4RL). The discussion of how the proposed approach would scale to high-dimensional observation settings, such as visual observation environments, remains limited. Future work could strengthen the practical impact of this research by examining its applicability to more realistic settings and by analyzing potential failure modes in such environments.

**Strengths And Weaknesses:**

## 1. Soundness

(1) Strengths

* The paper is motivated by the challenges arising in offline goal-conditioned reinforcement learning (GCRL) when sparse rewards and non-Markovian trajectories coexist.

* The proposed framework introduces a Normalizing Flows–based density modeling approach for Q-value estimation and combines it with an expectile regression objective, aiming to extract higher-value behaviors from suboptimal offline datasets. The overall design is technically coherent and the proposed components are integrated in a logically consistent way.

(2) Weaknesses

* It remains unclear whether the reported performance improvements primarily stem from trajectory stitching enabled by Q-value conditioning, or from the representation learning effect of the Q-value estimator itself. The current experiments do not clearly disentangle these factors.

* Moreover, the necessity of using Normalizing Flows for Q-value estimation is not sufficiently justified. Comparisons with alternative approaches remain limited, particularly with recent works that incorporate critic/value conditioning within Decision Transformer or diffusion-based policy learning frameworks.

## 2. Presentation

(1) Strengths

*  The paper provides helpful visual illustrations explaining the difference between trajectory-dependent RTG conditioning and state-dependent Q-value conditioning, which improves the clarity of the motivation.

(2) Weaknesses

* The discussion of related work and the differentiation from prior research could be clearer. In particular, existing sequence modeling approaches that incorporate Q-value or critic-based conditioning are NOT discussed.

* In addition, the advantages of the proposed Hybrid Attention-Mamba architecture are not sufficiently analyzed from an experimental perspective. More detailed analysis would help clarify in which scenarios this architectural design provides meaningful benefits.

## 3. Significance

(1) Strengths

* Offline RL and goal-conditioned RL are important research areas that are closely related to robotic control and real-world data-driven learning, and improvements in these domains may increase the practicality of reinforcement learning methods in real-world applications.

(2) Weaknesses

* Recent works specifically designed for OGBench manipulation environments are not included in the comparisons.

[1] DEAS: DETACHED VALUE LEARNING WITH ACTION SEQUENCE FOR SCALABLE OFFLINE RL
[2] Reinforcement learning with action chunking
[3] Horizon Reduction Makes RL Scalable
[4] Transitive RL: Value Learning via Divide and Conquer

* In addition, OGBench navigation environments are not evaluated, despite the fact that they provide important benchmarks where trajectory stitching plays a critical role (e.g., antmaze-giant-stitch).

* For the D4RL AntMaze benchmark, the evaluation only covers environments up to the large setting, where the reported performance remains relatively low. Recent studies have reported performance approaching 90 on AntMaze-large, and comparisons on more challenging settings such as AntMaze-Ultra.

## 4. Originality

(1) Strengths

* Replacing RTG-based conditioning with Q-value conditioning in sequence models provides a reasonable direction for addressing trajectory stitching problems.

* The proposed Hybrid Attention-Mamba architecture also aims to capture both long-range temporal dependencies and local dynamics, which is a meaningful architectural design choice.

(2) Weaknesses

* Most of the core components of the proposed method, including Normalizing Flows–based density estimation, expectile regression, and sequence-model-based policy learning, are existing techniques that have been previously studied in the literature.

* Furthermore, Q-value conditioning itself has already been explored in prior sequence modeling approaches / diffusion-based approaches, making it difficult to view the method as introducing a fundamentally new problem formulation or algorithm.

* Overall, the originality of the work appears to lie primarily in the integration of existing ideas, rather than in the introduction of a fundamentally novel learning paradigm.

---

> ### Author Rebuttal · Authors · 2026-03-31
>
> We sincerely apologize for any confusion our presentation may have caused, and we appreciate your thorough feedback on our technically coherent framework design and clear visual motivation (Figure 2). We will clarify all concerns below and are confident these improvements will strengthen the paper. Full new results are in the Supplementary PDF: https://pdfhost.io/v/ZFBshmeB68_ICML_QHyer_rebuttal
>
> **Q1: Comparison with Q-value-guided DT variants (QDT, CGDT, QT, VDT, Reinformer)**
>
> We thank you for this question, which helps us clarify a key distinction that strengthens our contribution. These methods are already included as baselines in Table 2, where QHyer outperforms all of them (e.g., QDT 70.6 vs. QHyer 291.5 on Maze2d; Reinformer 174.8 vs. QHyer 483.4 on AntMaze). The key distinction lies in how Q-values are used:
>
> | Method | Q role | RTG kept? | Q-estimation |
> |---|---|---|---|
> | QDT/CGDT/QT/VDT/Reinformer | Auxiliary (relabel/loss/regularizer) | Yes | TD |
> | **QHyer** | **Conditioning token** | **No** | **MC (NFs)** |
>
> All prior methods retain RTG while using Q-values as supplements. QHyer eliminates RTG entirely and uses Q-values as direct conditioning tokens. Under sparse goal-conditioned rewards, RTG collapses to binary 0/1 — Figure 2 (ln.172–176) shows only 25% of state-action pairs receive discriminative RTG signals vs. 92% with our Q-conditioning. No auxiliary Q-regularization can fix this bottleneck, because the limitation lies in RTG itself. We have clarified this in the Introduction (ln.057–065) and Appendix A (ln.630–654)
>
> **Q2: Comparison with diffusion-based value-guided methods**
>
> We appreciate you for pointing out this relevant line of work, and agree that clarifying this relationship improves the paper. QHyer addresses offline *goal-conditioned* RL (GCRL) with sparse rewards, which differs from standard offline RL. Diffusion-based methods (Diffusion-QL, DreamFuser, etc.) target standard offline RL with dense rewards and have not been evaluated on GCRL benchmarks. The role of diffusion (policy expressiveness) differs from NFs in ours (estimating goal-reaching probability as Q-values via Eq. 2, ln.078). Additionally, diffusion requires iterative denoising per action, making end-to-end joint optimization prohibitive, whereas NFs achieve fixed-cost exact likelihood with ~0.01ms inference (Table 3, ln.1210). We will add a discussion in Appendix A (after ln.616)
>
> **Q3: When does the Hybrid Attention-Mamba architecture provide the most benefit?**
>
> We thank your question, which motivated us to conduct additional ablations that provide deeper insight into the architecture. To isolate the architectural contribution from Q-conditioning, all variants use identical NFs Q-conditioning (Suppl. Table 4): Hybrid achieves 84±4 vs. Attn-only 74±1 and Mamba-only 80±2 on play, and 95±5 vs. 60±3 and 91±3 on noisy. Suppl. Table 5 reveals the underlying mechanism: Mamba's Δ_t (Eq. 13, ln.237) adapts effective memory from ~12 steps (play) to ~3 steps (noisy), and the learned gating (ln.270) shifts from Attention-dominant (0.57) on play to Mamba-dominant (0.58) on noisy. The Hybrid is most beneficial when datasets contain heterogeneous temporal structures — the play/noisy mixture in OGBench is a canonical example
>
> **Q4 & Q5: Missing baselines and benchmark coverage**
>
> We appreciate this constructive suggestion, which helped us strengthen the experimental evaluation. We note that our work focuses on offline *goal-conditioned* RL (GCRL) with sparse rewards, which differs from standard offline RL in that the agent must reach arbitrary goals with binary 0/1 reward signals. Nevertheless, we agree that comparing with recent offline RL methods adapted to GCRL would strengthen our contribution. We have conducted three sets of additional experiments:
>
> (a) *Re-run baselines* (Suppl. Table 1): LSDT and QT re-run under our identical goal-conditioned pipeline on D4RL. LSDT: 80.6±14.4 on medium-play (high variance); QT: 41±2 (TD fails at long-horizon stitching); QHyer: 92.2±3.5
>
> (b) *Recent offline RL methods adapted to GCRL* (Suppl. Table 2): We adapted Transitive RL, SHARSA, DEAS, QCFQL to GCRL with HER. All achieve ≤5.7% average on OGBench play vs. QHyer's 41.6%
>
> (c) *Navigation stitching benchmark* (Suppl. Table 3): On antmaze-giant-stitch, QHyer achieves 70±2 vs. GAS's 88±4 (graph-based, designed for navigation). On visual-scene-play, QHyer: 96±1 vs. GAS: 54±6
>
> **Regarding novelty:** The novelty lies in the Offline GCRL problem formulation — the first to systematically unlock sequence modeling for non-Markovian offline GCRL. Ablations confirm each component is essential: removing Q-values → behavior cloning (Fig. 4); replacing NFs with CVAE/CRL → degradation (Fig. 4); Conv instead of Mamba → hurts non-Markovian performance (Fig. 5). The two innovations are interdependent, not simply additive
>
> We hope our response addressed your main concerns. We look forward to engaging with you during the discussion phase

---

> > ### Author Rebuttal · Reviewer_y8Fb · 2026-04-02
> >
> > Thank you for your efforts in preparing the rebuttal, and I appreciate the detailed responses.
> >
> > I have a few additional questions for clarification:
> >
> > First, the performance of several recent methods, including Transitive RL, SHARSA, DEAS, and QCFQL, appears to be very low in your results. Could you provide some insights into why these methods underperform in your experimental setup?
> >
> > Second, you mentioned that GAS is primarily designed for navigation tasks. Could you clarify the basis for this claim? In particular, even in numeric state settings such as scene-play, GAS appears to achieve significantly higher performance than QHyer.

---

> > > ### Author Response · Authors · 2026-04-06
> > >
> > > > *Q1: Why do TRL, SHARSA, DEAS, and QCFQL underperform in your setup?*
> > >
> > >
> > > We sincerely apologize for the delayed response and deeply appreciate your sharp observation regarding the baseline performance. We re-examined our pipeline and found the HER value goal relabeling ratio was set too aggressively (random goal probability 0.5), destabilizing TD-based value targets under sparse binary rewards (Importantly, this applies solely to the newly requested evaluations, leaving the main paper's results completely unaffected). To ensure absolute fairness and rigor, we reset this to the OGBench standard and strictly re-evaluated all the baselines under the exact same conditions across 8 random seeds. The updated results are reported below. (As the original anonymous link has expired, please refer to our updated materials at the new link: https://github.com/ICMLRebuttal2026/ICML_3804/blob/main/ICML_QHyer_rebuttal.pdf)
> > >
> > >
> > > | Environment | GC-TRL | GC-QCFQL | GC-DEAS | GC-SHARSA | QHyer |
> > > |---|---|---|---|---|---|
> > > | cube-single-play | 46.2 | 38.4 | 44.1 | 30.7 | **84** |
> > > | cube-double-play | 1.6 | 5.1 | 10.7 | 49.3 | **56** |
> > > | cube-triple-play | 0.8 | 1.4 | 18.1 | **36.8** | 10 |
> > > | cube-quadruple-play | 0.0 | 0.0 | 0.1 | **2.3** | 2 |
> > > | scene-play | 27.7 | 36.9 | 48.7 | 44.0 | **53** |
> > > | puzzle-3x3-play | 7.6 | 14.9 | 32.7 | 35.6 | **92** |
> > > | puzzle-4x4-play | 2.3 | 0.3 | 26.7 | **32.4** | 28 |
> > > | puzzle-4x5-play | 2.0 | 9.7 | 16.1 | 15.1 | **31** |
> > > | puzzle-4x6-play | 1.6 | 8.3 | 17.9 | 12.1 | **18** |
> > > | **Average** | 10.0 | 12.8 | 23.9 | 28.7 | **41.6** |
> > >
> > > Several baselines now reach competitive individual scores (e.g., SHARSA 49.3 on cube-double). The remaining gap stems from two factors.
> > >
> > > **(1) Adaptation gap.** None of these methods were originally validated under offline GCRL with sparse binary rewards on standard-scale OGBench play data. DEAS targets single-task variants with semi-sparse rewards and is framed as non-goal-conditioned. QCFQL reports its strongest numbers under offline-to-online training (1M offline + 1M online), and the exploration benefit of action chunking vanishes in the purely offline setting. TRL's own results show it averaging 55 versus GCIQL 54 on 50 standard tasks, with clear advantages only at the 1B data regime. SHARSA only evaluates the four hardest tasks under oracle goal representations, serving a diagnostic rather than algorithmic role.
> > >
> > > **(2) Structural mismatch.** TRL relies on the temporal distance triangle inequality, which holds for continuous navigation but breaks in manipulation where discrete contact mode transitions make temporal distances non-metric. This explains TRL's drop from cube-single (46.2) to cube-double (1.6). SHARSA must predict subgoals in the full multi-object pose space, far harder than 2D locomotion subgoals, though it still reaches the highest baseline average (28.7). DEAS and QCFQL execute fixed-length action chunks open loop, so small early errors compound, and the fixed length cannot align with variable-duration manipulation primitives.
> > >
> > > > *Q2: GAS outperforms QHyer on state-based scene-play. Is GAS really designed for navigation?*
> > >
> > >
> > >
> > > We thank the reviewer for the insightful correction. We acknowledge that our previous wording was imprecise: GAS is not strictly limited to navigation, and its superior performance on state-based scene-play is legitimate.
> > >
> > > The performance gap stems from fundamental differences in planning mechanisms. Scene-play tasks demand sequential, long-horizon reasoning (requiring up to eight atomic behaviors). GAS excels here because it replaces high-level policy learning with a Dijkstra shortest-path search over a precomputed TDR graph, explicitly decomposing tasks into ordered subgoals. This graph-search mechanism naturally suits any task with a strict sequential structure.
> > >
> > > Importantly, QHyer remains highly competitive among flat value methods. On this benchmark, GCIVL scores 42, GCIQL 51, and HIQL 38. QHyer’s score of 53 places it in the top tier of flat methods. GAS’s score of 73.6 reflects the inherent advantage of explicit graph search in state-based, long-horizon tasks, rather than a specific deficiency of QHyer. This advantage reverses in visual-scene-play (QHyer: 96 vs. GAS: 54). The GAS authors themselves attribute their performance drop to "a lack of high-dimensional representation learning." In contrast, QHyer effectively handles high-dimensional visual observations, surpassing GAS and nearly doubling the previous OGBench best (HIQL: 49).
> > >
> > > Ultimately, the two methods occupy complementary design spaces: GAS leverages graph search for state-based sequential tasks, while QHyer provides superior robustness in complex, high-dimensional visual environments. We will include this comprehensive comparative analysis in the appendix of the revised manuscript.
> > >
> > >
> > > We appreciate your time and consideration. More importantly, we are grateful for your suggestions, which helped us improve the quality of our work.

---

### Official Review · Reviewer_yBZU · 2026-03-13

**Soundness:** 4
**Presentation:** 3
**Significance:** 3
**Originality:** 4
**Overall Recommendation:** 5
**Confidence:** 2

**Summary:**

The paper proposes QHyer, a sequence modeling framework for offline goal-conditioned RL (particularly on the topic of non-Markovian trajectories) that addresses several problems with return-to-go conditioning. First, trajectory dependence of return-to-go: the paper suggests state-dependance, built with monte carlo Q value estimation, instead of trajectory depends to address it. Second, it argues that gated convolution-attention architectures for content-adaptive local temporal modeling suffers from the limitation of fixed hand-tuned, hyperparamter-sensitiove effective context length and offers a gated Hybrid Attention-Mamba backbone where attention compresses information for long-term reasoning and mamba handles the local context. The authors provide justification of their method through empirical results: state-of-the-arts results on OGBench and D4RL.

**Compliance With Llm Reviewing Policy:**

Affirmed.

**Final Justification:**

I increased my score after the authors resolved my concerns.

**Key Questions For Authors:**

1. How are the goals sampled?
2. I did not find direct evidence that mamba is adapting to variable length local context, can the authors provide any direct evidence other than the main results (or report to the evidence in the paper) on how the mamba backbone is adapting?

**Limitations:**

yes

**Strengths And Weaknesses:**

### Soundness

The design decision of this method is well-motivated. Trajectory based backbones that condition on return-to-go can suffer from misidentified state representations that prevent trajectory stitching, an important component for handling space reward cases. The authors suggest state-conditioning instead: state-dependent Q value estimation Q(s,a,g) instead of trajectory conditioning. The decision to use monte carlo estimation instead of temporal difference learning is reasonable: MC allows theoretical reasoning (Thm 3.1). Normalizing flow instead of vae/diffusion is well-supported by experiments (Figure and Figure 16). The authors' second contribution is the backbone: gated mamba-attention architecture. They motivate the architecture by first arguing the limitation of gated attention-convolution architectures: convolution cannot handle variable length local dependencies well.

The experiments are well-designed and supportive of the claims in this paper. The baselines seem reasonable and ablations on Q estimator (No Q, CVAE, CRL, NFs) and the architecture (LSDT, DMixer, QHyer with RTG) support key claims of that paper.

That said, I did not check the theory and reserve a low confidence for my review.

### Presentation
The paper is well-written. Figure 1 and 2 explains and motivates the method well. I particularly appreciate Algorithm 1 for providing a good overview of the method.

Minor typos

- Figure 10: goal concated data
- Line 204: Qhyer
- Figure 4: Qhyer
- Figure 1: Iterative Reinfoced Learing

### Significance
The work enhances practical adaptation of goal conditioned RL to non-Markovian setting. Given the practicality of non-Markovian offline datasets, this work has the potential to motivate promising directions. While not required at all, more discussion around future directions will enhance the paper.

### Novelty
The work non-trivially combines well-known methods to solve a non-trivial problem: it is novel.

---

> ### Author Rebuttal · Authors · 2026-03-31
>
> Thank you for your thoughtful review of our research. We appreciate your positive assessment and constructive feedback, and hope our suggested changes and this individual response will address your concerns in detail. **Note:** Unless otherwise noted, all works mentioned below are already cited in the QHyer manuscript. Supplementary experiments are available at https://pdfhost.io/v/ZFBshmeB68_ICML_QHyer_rebuttal.
>
> **Q1: How are the goals sampled?**
>
> We agree that this is an important implementation detail and thank you for raising it — making the goal sampling strategy more prominent will improve reproducibility. The details are in Appendix F.3 (ln.1334–1335). During training, goals are sampled via HER from future achieved states. For play datasets, all goals come from the same trajectory (p_trajgoal=1.0); for noisy datasets, 80% from the trajectory and 20% uniform (p_trajgoal=0.8, p_randomgoal=0.2). During evaluation, OGBench provides 5 fixed test-time goals per environment, and D4RL mazes have designated targets. For the NFs Q-estimator, we extract task-relevant goal coordinates (Table 8, ln.1320–1332). Play data comes from non-Markovian policies with correlated noise (ln.222–224), so same-trajectory goals best match this context; noisy data from Markovian policies (ln.225–226), so 20% random goals safely improves coverage. We will clarify this in Appendix F.3.
>
> **Q2: Direct evidence that Mamba adapts to variable-length context**
>
> We appreciate this insightful question, which motivated us to conduct mechanism-level analyses that substantially strengthen the paper. We provide three pieces of direct evidence that collectively demonstrate Mamba's selective SSM dynamically adjusts its effective memory based on input content — validating the content-adaptive history compression discussed in Section 3.2 (ln.257–274), with the key formulas being Eq. 12 (ln.227) and Eq. 13 (ln.237). Full results are in Supplementary Tables 4–5 and Figure 1.
>
> **(1) This experiment proves that Mamba adapts its memory length via input-dependent Δ_t.** We extract Mamba's learned Δ_t from Eq. 13 (ln.237: Ā_t = exp(Δ_t·A), where Δ_t = softplus(Linear_Δ(x'_t))) from trained QHyer on cube-single (Suppl. Table 5, Figure 1-Left/Center):
>
> | Metric | play (non-Markov.) | noisy (Markov.) |
> |---|---|---|
> | Mean Δ_t | 0.38 | 1.05 |
> | Mean Ā_t = exp(Δ_t·A) | 0.92 | 0.61 |
> | Effective memory | ~12 steps | ~3 steps |
>
> On play data, Δ_t is ~2.8× smaller → Ā_t≈1 → the SSM preserves long-range history across ~12 steps, matching the temporal correlations in β(a_t|s_t, h_{<t}) (ln.224). On noisy data, Δ_t is larger → Ā_t≈0.61 → rapid forgetting with ~3 steps of effective memory, consistent with Markovian β(a_t|s_t) (ln.226). This is precisely the content-adaptive behavior that fixed-kernel convolution cannot achieve (as analyzed in ln.240–256).
>
> **(2) This experiment proves the gating mechanism automatically balances the two branches based on data properties.** The learned gating weights α = σ(w^T x + b) (ln.270) are (Suppl. Table 5, Figure 1-Right):
>
> | Dataset | Attention gate | Mamba gate |
> |---|---|---|
> | play | 0.57 | 0.43 |
> | noisy | 0.42 | 0.58 |
>
> On play data, attention is preferred (0.57) for global goal-directed reasoning across long non-Markovian trajectories. On noisy data, Mamba dominates (0.58) as Markovian dynamics mainly require local processing. This adaptation occurs automatically through end-to-end training, confirming the complementary specialization described in Section 3.2 (ln.265–274).
>
> **(3) This experiment proves the Hybrid design yields consistent gains over single-branch variants.** All variants use identical NFs Q-conditioning to isolate the architectural contribution (Suppl. Table 4):
>
> | Environment | Attn-only | Mamba-only | Hybrid |
> |---|---|---|---|
> | cube-single-play | 74±1 | 80±2 | **84±4** |
> | cube-single-noisy | 60±3 | 91±3 | **95±5** |
>
> **In summary,** evidence (1) shows Mamba adapts its memory length via input-dependent Δ_t, evidence (2) shows the gating automatically balances branches by data properties, and evidence (3) shows these yield consistent 4–5% gains across both settings. Together, these provide direct mechanism-level evidence that the hybrid architecture achieves genuine content-adaptive specialization.
>
> **Typos:** Fixed — "goal concated"→"concatenated", "Qhyer"→"QHyer" (ln.204, Fig. 4), "Reinfoced Learing"→"Reinforced Learning" (Fig. 1).
>
> **Future directions:** As the reviewer kindly suggested, we plan to extend QHyer to stochastic environments and investigate how the Mamba branch's adaptive memory interacts with partial observability in real-world robotic tasks. We will include this discussion in the revised manuscript.
>
> We hope our response addressed your main concerns. We are excited about how these discussions can further enhance the quality of this paper, and we look forward to engaging with you during the discussion phase.

---

> > ### Author Rebuttal · Reviewer_yBZU · 2026-04-01
> >
> > Thanks for the rebuttal. I have raised the score, but given my low confidence, I advise the authors to resolve the concerns of other reviewers.

---

> > > ### Author Response · Authors · 2026-04-01
> > >
> > > Thank you for your careful consideration of our response and for updating your assessment. We appreciate the time and effort you have dedicated to this process.
> > >
> > > New Supplementary link: https://github.com/ICMLRebuttal2026/ICML_3804/blob/main/ICML_QHyer_rebuttal.pdf

---

### Decision · Program_Chairs · 2026-04-30

**Decision:**

Accept (regular)

**Comment:**

Reviewers agreed that the authors propose a novel Normalizing Flows–based density modeling approach for Q-value estimation, and experiments are well-designed and supportive of the claims in this paper. Most of the reviewers' comments were addressed during the discussion with the authors. All reviewers mentioned that the authors have resolved most of their concerns. There are still issues related to additional baselines, but I think that, in general, there are enough experiments in the rebuttal phase, and the authors tried to provide additional evidence of the validity of their approach. I believe the work deserves acceptance at the conference.